# Selective Concept Bottleneck Models Without Predefined Concepts

**Simon Schrodi**[*]                                                    *schrodi@cs.uni-freiburg.de*
*University of Freiburg*

**Julian Schur**[*]                                                     *julian.schur@student.kit.edu*
*University of Freiburg, Karlsruhe Institute of Technology*

**Max Argus**                                                          *argusm@cs.uni-freiburg.de*
*University of Freiburg*

**Thomas Brox**                                                        *brox@cs.uni-freiburg.de*
*University of Freiburg*

**Reviewed on OpenReview:** *https: // openreview. net/ forum? id= PMO3OTLI4l*

## Abstract

Concept-based models like Concept Bottleneck Models (CBMs) have garnered significant interest for improving model interpretability by first predicting human-understandable concepts before mapping them to the output classes. Early approaches required costly concept annotations. To alleviate this, recent methods utilized large language models to automatically generate class-specific concept descriptions and learned mappings from a pretrained black-box model's raw features to these concepts using vision-language models. However, these approaches assume prior knowledge of which concepts the black-box model has learned. In this work, we discover the concepts encoded by the model through unsupervised concept discovery techniques instead. We further leverage a simple input-dependent concept selection mechanism that dynamically retains a sparse set of relevant concepts of each input, enhancing both sparsity and interpretability. Our approach not only improves downstream performance, but also needs significantly fewer concepts for accurate classification. Lastly, we show how large vision-language models can guide the editing of our models' weights to correct model errors.

## 1 Introduction

Deep neural networks have achieved tremendous success in a variety of tasks on various input modalities. However, they are *black-box* models, making it difficult for humans to understand and comprehend their decisions. Thus, there has been considerable recent interest in developing *interpretable* models. One popular framework is Concept Bottleneck Models (CBMs) (Koh et al., 2020), i.e., models that first predict human-understandable concepts and then use these concepts to predict the classes (Lampert et al., 2009; Kumar et al., 2009). Initial CBMs are trained in an end-to-end fashion through supervision on *both* the concepts and classes. However, the need for human-annotated concepts during model training requires the time-consuming and expensive collection of such.

To address this limitation of initial CBMs, recent work (Yuksekgonul et al., 2023; Oikarinen et al., 2023; Menon & Vondrick, 2023; Laguna et al., 2024; Dominici et al., 2024) has proposed converting pretrained black-box models into CBMs in a *post-hoc* fashion. To avoid the need for annotations, they leveraged large language models (e.g., GPT-3 (Brown et al., 2020)) to generate class-specific language descriptions and learned a mapping from the black-box model's uninterpretable features to these concepts using vision-language models (e.g., CLIP (Radford et al., 2021)). However, this raises a crucial question:

---

[*]Equal contribution.

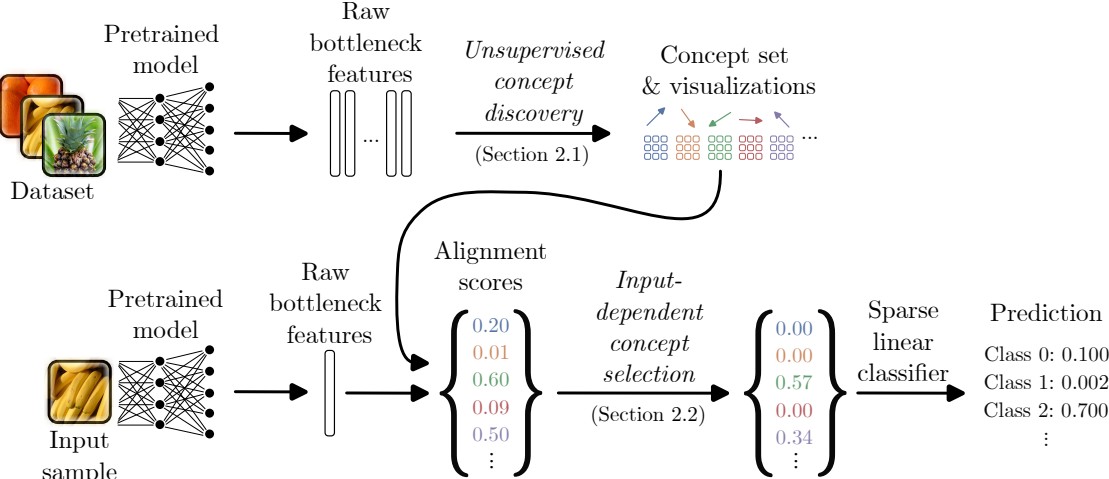

Figure 1: **Overview of Unsupervised Concept Bottleneck Models (UCBMs).** Top: We propose to extract concepts from raw bottleneck features of a pretrained black-box model using an unsupervised concept discovery method (Section 2.1). Bottom: We compute the alignment between the bottleneck's features and previously discovered concepts (middle). Finally, we train an interpretable classifier consisting of our proposed input-dependent concept selection mechanism and a sparse linear classifier (middle to right, Section 2.2).

*How can we know **a priori** which concepts a pretrained black-box model has learned?*

Instead of defining the concepts in advance, we propose to discover concepts that accurately decompose the features learned by the black-box model. To do so, we draw from the rich literature on unsupervised concept discovery (Ghorbani et al., 2019; Zhang et al., 2021; Zou et al., 2023; Fel et al., 2023b; Vielhaben et al., 2023; Fel et al., 2023a; Huben et al., 2024; Stein et al., 2024). We chose CRAFT (Fel et al., 2023b) for our experiments because it has been shown to yield human-understandable concepts (Fel et al., 2023a), but other techniques are also possible. CRAFT employs non-negative matrix factorization (Lee & Seung, 1999) to decompose each feature activation into a sparse linear combination of concept vectors. The set of shared concept vectors forms a dictionary matrix. After learning this dictionary matrix, we compute the alignment between the raw bottleneck features and the concept vectors to measure a concept's presence or absence.

Subsequently, we train an interpretable linear classifier on the concepts' alignment scores, linking the alignment scores to the predictions. Previous work (Yuksekgonul et al., 2023; Oikarinen et al., 2023; Srivastava et al., 2024) has shown that a sparsity penalty on the linear classifier's weights ensures that each class relies on only a sparse set of concepts. However, they did not examine the per-sample number of concepts that affect the classification across *all* classes. That is, while individual classes rely on sparse sets of concepts, the overall model depends on substantially more. Empirically, we found that typically 90% of the available concepts—up to ca. 4200 concepts (see Table 1)—affect the classification per input. As a result, it complicates the interpretation of the model's classification.

To address these challenges, we propose an *input-dependent concept selection mechanism* that ensures that only a sparse set of concepts relevant for the classification of an individual input sample is dynamically retained. We achieve this by applying a non-linear function before the sparse linear classifier to filter out (i.e., zero out) concepts. We enforce the filtering either by forcing the output of the non-linear function to be sparse or by directly controlling its sparsity through its hyperparameter. In our experiments, the TopK function (Makhzani & Frey, 2014) performed best. This mechanism allows the concepts that are retained or removed to vary between inputs, making it input-dependent. Importantly, it also preserves the interpretability of CBMs, as the predictions remain linear w.r.t. the retained concepts. Finally, we show that it also effectively controls information leakage; a common problem of CBMs (Mahinpei et al., 2021; Yan et al., 2023; Srivastava et al., 2024).

In summary, our contributions are as follows:

- We propose a new type of CBM called Unsupervised Concept Bottleneck Models (UCBMs)[1]; see Figure 1 for an overview. UCBMs convert pretrained, black-box models into a CBM by *discovering and using the concepts that the black-box model has learned.*

- We propose an input-dependent concept selection mechanism that *dynamically retains a sparse set of concepts* relevant to classification. For example, as few as ca. 1.4% of the available concepts are used per input (Table 1).

- We show that UCBMs improve *performance* while having a substantially higher degree of *sparsity* compared to previous work (Figure 3) and effectively controls information leakage (Figure 5).

- We show that UCBMs are *interpretable* qualitatively and through a user study (Section 3.2), and show that large vision-language models can help us to *intervene* on UCBMs' weights to fix errors (Section 3.3).

## 2 Unsupervised Concept Bottleneck Models with input-dependent concept selection

In this section, we introduce Unsupervised Concept Bottleneck Models (UCBMs), a novel CBM that uses concepts that are automatically discovered and most accurately decompose the features learned by a black-box model (Section 2.1), dynamically only retains the concepts most relevant to classification of each input, and finally classifies the input with a sparse linear model (Section 2.2). Figure 1 provides an overview of our method, and the above steps are described in detail below.

**Notations.** Let $f : \mathcal{X} \to \mathbb{R}^p$ be a pretrained, black-box model's feature extractor that maps from an input space $\mathcal{X} \subseteq \mathbb{R}^d$ to the bottleneck feature space of a size of $p$. Further, let $\mathbf{X} \in \mathbb{R}^{N \times d}$ be the input data matrix where the $i^{th}$ row is the input $\mathbf{x}_i \in \mathbf{X}$ and let $\mathbf{A} = f(\mathbf{X}) \in \mathbb{R}^{N \times p}$ be the bottleneck feature activations. Lastly, let $\mathcal{Y}$ denote the class label space.

### 2.1 Discovery of concepts learned by the black-box model

Previous post-hoc CBMs have either used human-annotated concepts (Yuksekgonul et al., 2023; Laguna et al., 2024; Dominici et al., 2024) or aligned the black-box model's features with precomputed text features from vision-language models, using natural language descriptions, such as those generated by a large language model (Yuksekgonul et al., 2023; Oikarinen et al., 2023; Menon & Vondrick, 2023; Laguna et al., 2024). Importantly, both approaches rely on a *predefined set of concepts*—either through concept annotations or language descriptions thereof—implicitly assuming which concepts the black-box model has learned. However, the concepts are typically unknown in advance.

**Discovering the concepts that the black-box model has learned.** To address this, we propose using unsupervised concept discovery techniques for UCBMs. These enable us to discover the concepts that the black-box model has actually learned, and do not require defining the concepts in advance.

Formally, the goal of unsupervised concept discovery is to extract a small set of interpretable concepts $\mathbf{C}$ that most faithfully reconstruct the feature activations $\mathbf{A}$. Assuming linearity of concepts, as per the superposition hypothesis (Kim et al., 2018; Elhage et al., 2022), unsupervised discovery methods can be understood as an instance of a dictionary learning problem (Dumitrescu & Irofti, 2018):

$$(\mathbf{U}^*, \mathbf{C}^*) = \underset{\mathbf{U}, \mathbf{C}}{\arg\min} \ ||\mathbf{A} - \mathbf{U}\mathbf{C}||_F^2 \qquad , \qquad (1)$$

where $\mathbf{U} \in \mathbb{R}^{N \times |\mathbf{C}|}$ (sparse coefficient matrix) represents the activations $\mathbf{A} = f(\mathbf{X}) \in \mathbb{R}^{N \times p}$ w.r.t. a new basis spanned by the set of $|\mathbf{C}|$ concept activation vectors $\mathbf{C} \in \mathbb{R}^{|\mathbf{C}| \times p}$ (dictionary matrix), and $||\cdot||_F$ denotes the Frobenius norm. Intuitively, we learn a sparse linear decomposition of the feature activations of each input in Equation 1, where we weigh the shared concept vectors by the input-specific sparse coefficients. Fel et al. (2023a) showed that previous methods, such as K-Means (Ghorbani et al., 2019), PCA (Zhang et al., 2021; Zou et al., 2023), non-negative matrix factorization (Lee & Seung, 1999; Olah et al., 2018; Zhang et al.,

---

[1] Code is available at https://github.com/lmb-freiburg/ucbm.

2021; McGrath et al., 2022; Fel et al., 2023b), or sparse autoencoders (Makhzani & Frey, 2014; Huben et al., 2024), only differ in their constraints on $\mathbf{U}, \mathbf{C}$ in Equation 1.

In this work, we chose non-negative matrix factorization (i.e., CRAFT (Fel et al., 2023b)) for UCBMs, as it has been shown to discover human-understandable concepts (Fel et al., 2023a). However, we emphasize that UCBMs will benefit from future unsupervised concept discovery methods.

## 2.2 Learning the classifier with input-dependent concept selection

In the previous subsection, we discovered concept vectors $\mathbf{c}_j$ that most accurately decompose the uninterpretable features of a black-box model. Next, we compute the alignment scores between each concept vector and the model's features, denoted as $\text{sim}_{\mathbf{C}}(\mathbf{x}_i) \in [-1, 1]^{|\mathbf{C}|}$, where $\text{sim}_{\mathbf{C}}(\mathbf{x}_i)_j := \frac{\langle \mathbf{a}_i, \mathbf{c}_j \rangle}{||\mathbf{a}_i||_2 \cdot ||\mathbf{c}_j||_2}$ is the cosine similarity between the feature activations $f(\mathbf{x}_i) = \mathbf{a}_i$ of input $\mathbf{x}_i$ and concept $\mathbf{c}_j \in \mathbf{C}$. Then, we dynamically select the most relevant concepts and subsequently classify the input with a sparse linear model (Wong et al., 2021). Both are described in detail below.

**Sparse linear classifier.** Following Yuksekgonul et al. (2023); Oikarinen et al. (2023); Srivastava et al. (2024), we learn a sparse linear classifier by enforcing sparsity on its weight matrix (Wong et al., 2021):

$$\min_{\mathbf{W},\mathbf{b}} \sum_{i=1}^{N} \mathcal{L}(\mathbf{W}\text{sim}_{\mathbf{C}}(\mathbf{x}_i) + \mathbf{b}, y_i) + \lambda_w \underbrace{R_\alpha(\mathbf{W})}_{\mathcal{L}_{\text{sparsity}}^{\mathbf{W}}} \quad , \tag{2}$$

where $\mathbf{W} \in \mathbb{R}^{|\mathcal{Y}| \times |\mathbf{C}|}$ are the weights, $\mathbf{b} \in \mathbb{R}^{|\mathcal{Y}|}$ is the bias, $y_i \in \mathcal{Y}$ is the target class for input $\mathbf{x}_i$, $\mathcal{L}$ represents the task-specific loss function (cross-entropy loss throughout this work), $\lambda_w$ controls the regularization strength on $\mathbf{W}$, and $R_\alpha(\mathbf{W}) := (1 - \alpha)\frac{1}{2}||\mathbf{W}||_F + \alpha||\mathbf{W}||_{1,1}$ denotes the elastic net regularization (Zou & Hastie, 2005). Note that $\text{sim}_{\mathbf{C}}(\mathbf{x}_i)$ is normalized and frozen during optimization. Importantly, the sparsity aims to make the linear model's classifications sparse and Yuksekgonul et al.; Oikarinen et al. & Srivastava et al. have shown that *an individual class* indeed relies on only a sparse set of concepts.

The main limitations with only applying sparsity on the weights $\mathbf{W}$ are that it fails to produce *globally sparse* classifications and is input-independent. This lack of (global) sparsity limits interpretability and makes it challenging to comprehend a prediction. Specifically, we found that even when a concept is non-visible, it impacts classification—either for the predicted class or any other class (Table 1). We consider a concept to be actively contributing if it has a non-zero influence on the output (see Equation 7 for details). The reason that the concepts are non-zero and, consequently, influence classification is that the cosine similarities between the black-box model's activations and concepts are generally non-zero.[2]

**Input-dependent concept selection mechanism.** To ensure that only few concepts affect classification per input without significant performance sacrifices, we propose a simple yet effective *input-dependent concept selection mechanism*. Specifically, we introduce a concept selector $\pi : \mathbb{R}^{|\mathbf{C}|} \to \mathbb{R}^{|\mathbf{C}|}$, which takes the alignment scores $\text{sim}_{\mathbf{C}}(\mathbf{x}_i)$ as input and outputs a sparse set of non-zero (i.e., active) scores and zeroes out the others. We enforce sparsity through a penalty term on concept selector's output: $\mathcal{L}_{\text{sparsity}}^{\pi} = ||\pi(\cdot)||_0$. Intuitively, the sparsity penalty $\mathcal{L}_{\text{sparsity}}^{\pi}$ drives the concept selector $\pi$ to only retain a sparse set of concepts which are important for classifying the input $\mathbf{x}_i$, as signaled by the task-specific loss $\mathcal{L}$ in Equation 2.

We considered three candidates for the implementation of the input-dependent concept selection mechanism (please refer to Appendix C for further technical details):

- **ReLU:** We define the concept selector using the ReLU activation function as:

$$\pi(\mathbf{x}_i) := \max(0, \text{sim}_{\mathbf{C}}(\mathbf{x}_i) - \mathbf{o}) \text{ with trainable offset parameter } \mathbf{o} \in \mathbb{R}_+^{|\mathbf{C}|} \quad . \tag{3}$$

  We apply elastic net regularization on the selector's output: $\mathcal{L}_{\text{sparsity}}^{\pi} = R_\alpha(\pi(\mathbf{x}_i))$.

---

[2]While the classifier could technically "turn off" a concept $\mathbf{c}_j$ by setting its associated column vector to the null vector ($\mathbf{W}_{:,j} = 0$), this would effectively reduce the number of concepts and degrades performance, e.g., see Figure 4. Consequently, the sparse linear classifier is unlikely to learn many of such null vectors.

- **JumpReLU:** We use JumpReLU activation function (Erichson et al., 2019) for concept selection with trainable offset parameter $\mathbf{o} \in \mathbb{R}_+^{|\mathbf{C}|}$ and the Heaviside step function $H$. We define the concept selector as:

$$\pi(\mathbf{x}_i) := \mathrm{sim}_{\mathbf{C}}(\mathbf{x}_i) \cdot H(\mathrm{sim}_{\mathbf{C}}(\mathbf{x}_i) - \mathbf{o}) = \begin{cases} 0, & \mathrm{sim}_{\mathbf{C}}(\mathbf{x}_i) \leq \mathbf{o} \\ \mathrm{sim}_{\mathbf{C}}(\mathbf{x}_i), & \mathrm{sim}_{\mathbf{C}}(\mathbf{x}_i) > \mathbf{o} \end{cases} \quad . \tag{4}$$

Following Rajamanoharan et al. (2024), we compute the gradients of the *expected* loss using straight-through-estimators (Bengio et al., 2013). We use the following sparsity penalty $\mathcal{L}_{\mathrm{sparsity}}^{\pi} = \sum_j^{|\mathbf{C}|} H(\mathrm{sim}_{\mathbf{C}}(\mathbf{x}_i)_j - \mathbf{o}_j)$. Note that $\mathcal{L}_{\mathrm{sparsity}}^{\pi}$ directly optimizes L0.

- **TopK:** The TopK activation function (Makhzani & Frey, 2014) only keeps the $k \ll |\mathbf{C}|$ concepts with the largest alignment scores and zeroes out the remaining concepts:

$$\pi(\mathbf{x}_i) := \mathrm{TopK}_k(\mathrm{sim}_{\mathbf{C}}(\mathbf{x}_i) - \mathbf{o}) \text{ with trainable offset parameter } \mathbf{o} \in \mathbb{R}_+^{|\mathbf{C}|} \quad . \tag{5}$$

Note that the sparsity can be directly controlled by $k$ and, thus, $\mathcal{L}_{\mathrm{sparsity}}^{\pi} = 0$.

**Final interpretable classifier.** We obtain the final interpretable classifier by plugging Equation 3, 4, or 5 into Equation 2 together with the respective implementation of $\pi$ and $\mathcal{L}_{\mathrm{sparsity}}^{\pi}$:

$$\min_{\mathbf{W}, \mathbf{b}, \mathbf{o}} \sum_{i=1}^{N} \mathcal{L}(\mathbf{W}\pi(\mathbf{x}_i) + \mathbf{b}, y_i) + \lambda_w \mathcal{L}_{\mathrm{sparsity}}^{\mathbf{W}} + \lambda_\pi \mathcal{L}_{\mathrm{sparsity}}^{\pi} \quad , \tag{6}$$

where $\lambda_\pi$ (or $k$ for TopK) controls the regularization strength of $\mathcal{L}_{\mathrm{sparsity}}^{\pi}$. Appendix C provides a detailed overview of all variants. It is important to note that the selection of concepts is learned in an unsupervised manner, and that the prediction remains linear w.r.t. the *active* concepts ($\pi(\mathbf{x}_i) \neq 0$).

**Concept dropout.** During initial experiments, we found that models became overly reliant on a single concept. To reduce this reliance, we added a dropout layer (Srivastava et al., 2014) after concept selection. As dropout is applied per concept, it encourages the model to spread its classification decisions across more concepts. Interestingly, we found that this could also improve performance.

## 3 Experiments

We evaluated UCBM on diverse image classification tasks and compared it to relevant baselines. We show that UCBMs outperform prior work and narrow the gap to their black-box counterparts, while relying on substantially fewer concepts globally in their classification (Section 3.1). Then, we demonstrate the interpretability qualitatively as well as through a user study (Section 3.2). Lastly, we showcase how large vision-language models can be leveraged to intervene on UCBMs by informing weight editing in order to fix model errors (Section 3.3). Appendix N provides further analysis on the out-of-distribution robustness, fairness, and shape vs. texture bias of UCBMs.

**Datasets & black-box feature backbones.** The CBMs are evaluated on ImageNet (Deng et al., 2009) with a pretrained ResNet-50 V2 (He et al., 2016), CUB (Wah et al., 2011) with ResNet-18 pretrained on CUB, and Places-365 (Zhou et al., 2017) with ResNet-18 pretrained on Places-365.[3] These datasets cover a diverse set of tasks from standard image classification (ImageNet), fine-grained classification (CUB), to scene recognition (Places-365). Experiments with Inception and transformer feature backbone are done in Appendix F. We find that UCBMs achieve performance close to the original black-box models, consistent with the results observed for the ResNet feature backbones in Table 2.

**Implementation details.** We trained our UCBMs with Adam (Kingma & Ba, 2015) and cosine annealing learning rate scheduling (Loshchilov & Hutter, 2017) for 20 epochs. We used a learning rate of 0.001 on ImageNet and Places-365, and 0.01 on CUB; except for the JumpReLU for which we set it to 0.08 on CUB. We set $\alpha = 0.99$ for the elastic net regularization for all variants. We tuned the other hyperparameters ($\lambda_\pi$ or $k$, $\lambda_w$, and dropout rate) to yield a good trade-off between performance, sparsity, and fair comparability. Refer to Appendix D for the hyperparameters and to Figure 6 and Appendix G for their effect.

---

[3]Models are provided at https://github.com/pytorch/vision (ImageNet), https://github.com/osmr/imgclsmob (CUB), and https://github.com/Trustworthy-ML-Lab/Label-free-CBM (Places-365).

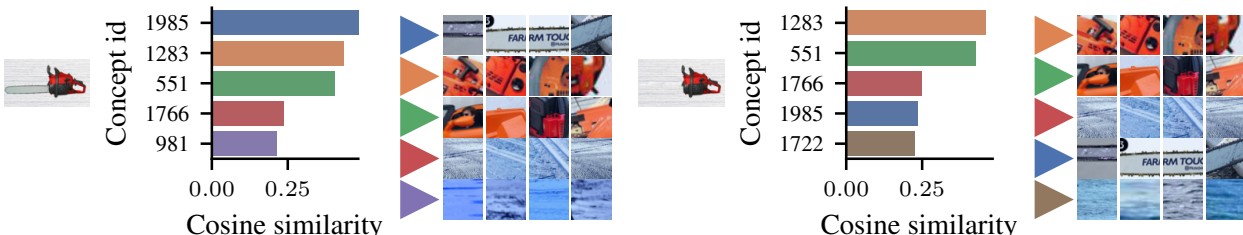

Figure 2: **The discovered concepts exhibit faithful behavior.** Removing the saw blade (right) from the original image (left) shrinks the alignment score of the respective concept 1985 (blue). Concepts are represented by their most activating crops. Additional results are provided in Appendix A.

**Experimental setup.** Since the number of concepts $|\mathbf{C}|$ substantially influence downstream performance (see Figure 4), we set $|\mathbf{C}|$ proportional to the number of classes with various (expansion) factors $\{0.5, 1, 3, 5\}$. All models were trained on a single NVIDIA RTX 2080 GPU and a full training run took from few minutes to a maximum of 1–2 days depending on dataset size and number of concepts $|\mathbf{C}|$. We report top-1 accuracy on the standard holdout sets throughout our experiments.

**Baselines.** We compared our UCBMs to Post-hoc CBM (Yuksekgonul et al., 2023), Label-free CBM (Oikarinen et al., 2023), and VLG-CBM (with NEC = 5) (Srivastava et al., 2024), as they are the most related to our work and the latter is the current state-of-the-art CBM. Note that Post-hoc CBM requires concept annotations and is therefore not applicable on ImageNet and Places-365. Finally, we compared our concept selectors with the binary (latent) indicator concept selector proposed by Panousis et al. (2023). We reproduced the baseline results using their respective original codebases.

**Quality of the discovered concepts.** Before we evaluated UCBMs, we verified that the discovered concepts behave faithfully. For this, we analyzed the change in cosine similarities between feature activations and concepts after the removal of relevant image parts of a certain concept; see Figure 2 and Appendix A. For example, as we remove the saw blade (concept 1985), the cosine similarity of the aforementioned concept decreases from ca. 0.5 to around 0.25 (Figure 2). We also manually verified that concepts are semantically consistent and human-understandable. The quality can be exemplarily seen in the top activating crops (i.e., the top-n crops are selected based on the cosine similarity of their bottleneck feature activations to the concept) throughout this paper. In addition, we evaluated the degree of polysemanticity in Appendix B.

## 3.1 Sparsity and performance results

**How sparse are UCBMs' decisions?** Previous work evaluated sparsity based on (the average per-class) number of non-zero weights in $\mathbf{W}$ (Oikarinen et al., 2023; Srivastava et al., 2024). However, these approaches fail to consider two important factors: (1) how many concepts influence classification across all inputs (globally), and (2) that certain concepts can be inactive for specific inputs, e.g., their value is zero.

To account for the aforementioned, we propose to compute the average number of concepts that actively influence the classification decision for each input $\mathbf{x}_i$. Formally, we consider concept $\mathbf{c}_j$ active for the classification of input $\mathbf{x}_i$ if

$$\underbrace{\mathbf{c}_j = \pi(\mathbf{x}_i)_j \neq 0}_{\text{is the concept } \mathbf{c}_j \text{ active?}} \quad \wedge \quad \underbrace{\exists\, y_i \in \{1, ..., |\mathcal{Y}|\} \text{ for which } \mathbf{W}_{y_i, j} \neq 0}_{\text{does the concept } \mathbf{c}_j \text{ have an effect on any class?}} \quad . \tag{7}$$

Tables 1 and 5 show that UCBMs with concept selection use substantially fewer concepts than UCBM without concept selection and the other baselines. For example, on ImageNet, UCBM with TopK concept selector uses an average of 42.0 concepts per input, while Label-free CBM, VLG-CBM, UCBM with binary indicator concept selection, and UCBM without concept selection use averages of 4238.0, 3018.97, 1995.7, or 3000.0, respectively. We find similar differences for CUB and Places-365.

**How good is the performance of UCBMs?** Table 2 shows that UCBMs mostly outperform the baseline methods across all datasets, while being substantially sparser (Tables 1 and 5 and Figure 3). The

Table 1: **The concept selection mechanism leads to substantially fewer concepts being used in the classification.** We report the mean percentage number of active concepts according to Equation 7 w.r.t. to the total number of concepts $|\mathbf{C}|$. Absolute numbers are provided in Table 5 in Appendix E. Label-free CBM, VLG-CBM, UCBM without concept selection, and UCBM with binary indicator use many more concepts than our UCBM variants with concept selection.

| | Mean number of active concepts/$|\mathbf{C}|$ ($\downarrow$) | | |
|---|---|---|---|
| Method | ImageNet | CUB | Places-365 |
| Post-hoc CBM (Yuksekgonul et al., 2023) | n/a | 100% | n/a |
| Label-free CBM (Oikarinen et al., 2023) | 93.74% | 99.95% | 90.64% |
| VLG-CBM (Srivastava et al., 2024) | 70.21% | 98.66% | 63.27% |
| UCBM w/o concept selection | 100% | 100% | 100% |
| UCBM with binary indicator (Panousis et al., 2023) | 66.52% | 100% | 49.28% |
| UCBM with ReLU concept selector | 1.59% | **30.5%** | *8.9%* |
| UCBM with JumpReLU concept selector | *1.43%* | *31.15%* | 9.11% |
| UCBM with TopK concept selector | **1.4%** | 32.1% | **8.88%** |

Table 2: **UCBMs mostly outperform the baselines and reduce the gap to the original, black-box model.** We report mean top-1 accuracy with standard deviation across three training runs (we kept the discovered concepts fixed). Note that the methods use different levels of sparsity (see Table 1) and refer to Figure 3 that plots sparsity against performance.

| | | Top-1 test accuracy ($\uparrow$) | | |
|---|---|---|---|---|
| Method | Sparse? | ImageNet | CUB | Places-365 |
| Original, black-box model | ✗ | 80.9 | 76.7 | 53.69 |
| Post-hoc CBM (Yuksekgonul et al., 2023) | (✓) | n/a | 60.10 | n/a |
| Label-free CBM (Oikarinen et al., 2023) | (✓) | 78.09 | 74.38 | 50.67 |
| VLG-CBM (Srivastava et al., 2024) | (✓) | 78.78 | **75.44** | 51.67 |
| UCBM w/o concept selection | (✓) | **79.80 ± 0.027** | *75.15 ± 0.037* | **52.41 ± 0.028** |
| UCBM with binary indicator (Panousis et al., 2023) | (✓) | 77.42 ± 0.056 | 74.93 ± 0.309 | 50.91 ± 0.105 |
| UCBM with ReLU concept selector | ✓ | 79.07 ± 0.029 | 74.61 ± 0.128 | 50.86 ± 0.021 |
| UCBM with JumpReLU concept selector | ✓ | *79.49 ± 0.016* | 74.57 ± 0.290 | *51.24 ± 0.019* |
| UCBM with TopK concept selector | ✓ | 79.32 ± 0.009 | 74.96 ± 0.083 | 51.20 ± 0.050 |

performance-sparsity trade-off is visualized in Figure 3, where we control the number of active concepts (according to Equation 7) by varying the hyperparameter $k$ for UCBM with TopK concept selector, or $\lambda_\pi$ for UCBM with ReLU or JumpReLU concept selector. We find that some models that allow for more concepts (e.g., UCBMs without concept selection) unsurprisingly outperform our UCBM variants with concept selection. However, the UCBM variants with concept selection are substantially sparser and typically achieve at least competitive, but mostly superior, task performance to the baselines. For example, our UCBM variants outperform *all* baselines on ImageNet and *all but VLG-CBM* on CUB and Places-365. Besides that, Figure 3 shows that one can control the sparsity-performance trade-off through the respective hyperparameters. This allows practitioners to set these hyperparameters according to their desired balance between sparsity (and better interpretability) and performance, based on the requirements of their application.

**Effect of the total number of concepts C.** We found that performance is strongly influenced by the total number of concepts $|\mathbf{C}|$ used. In Figure 4, we varied the number of concepts to assess this and, as expected, find that increasing $|\mathbf{C}|$ improves performance. Note that UCBMs achieve competitive but mostly superior performance (Table 2 and Figure 3) while using a smaller number of concepts $|\mathbf{C}|$.

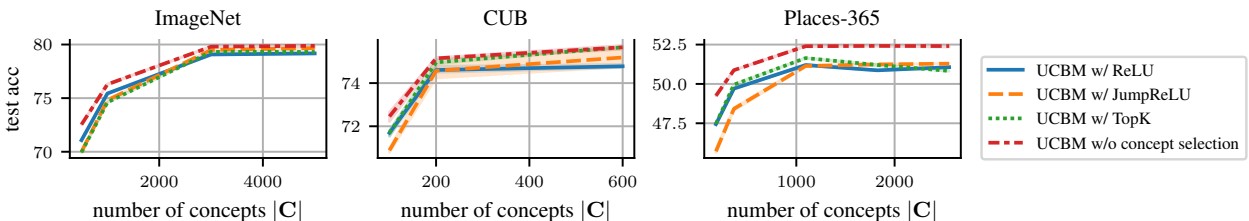

Figure 3: **Trade-off curves between sparsity and performance.** We plot the mean number of active concepts per input according to Equation 7 as we decrease $k$ (for TopK) or increase $\lambda_\pi$ (for the others). For UCBMs we plot the Pareto-curves. The UCBMs are substantially sparser than the baselines (see also Table 1). Our UCBMs Pareto-dominate all baselines on ImageNet and Places-365, while only being outperformed by VLG-CBM on CUB (though it has substantially higher sparsity).

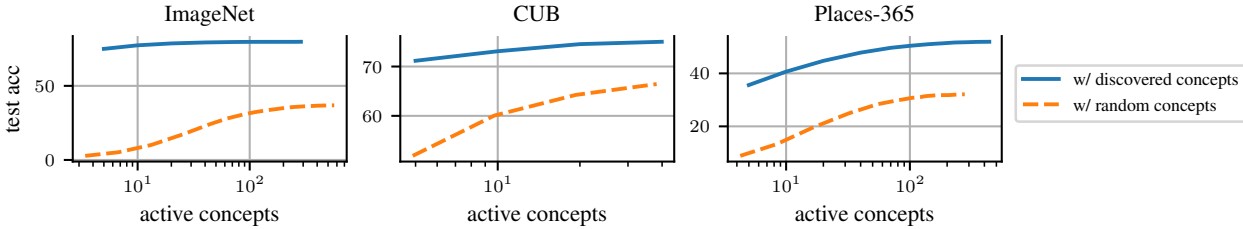

Figure 4: **The more concepts $|\mathbf{C}|$, the better UCBMs' performance.** We varied the total number of available concepts $|\mathbf{C}|$. As expected, the more available concepts $|\mathbf{C}|$, the better the performance.

Figure 5: **The TopK concept selector effectively controls information leakage**. Previous work showed that even using random concepts could yield strong CBMs, suggesting information leakage. However, performance with random concepts declines quickly, whereas it remains consistently high when using the discovered concepts when varying $k$ of the TopK concept selector.

**Concept selection effectively controls information leakage.** Recent work showed that CBMs' concept prediction may encode unintended class information (Margeloiu et al., 2021). For example, even many random concepts can achieve strong downstream performance (Yan et al., 2023; Midavaine et al., 2024; Srivastava et al., 2024). Figure 5 shows that $k$ effectively controls information leakage, as the performance of random concepts quickly drops when using smaller $k$ (fewer active concepts).

**Sensitivity analysis.** We varied $\lambda_w$ (Figure 6a), $k$ (Figure 6b), and dropout rate (Figure 6c) to analyze their impact on sparsity and performance. We find that only $k$ controls sparsity (Equation 7) in TopK, whereas for the other concept selectors, all hyperparameters affect sparsity (see Appendix G). We consider this is as an advantage of TopK, as it disentangles the effect of the hyperparameters. This is discussed in more detail in Appendix G. For performance, we find that larger $\lambda_w$ and smaller $k$ lead to worse performance. For dropout rate, there typically seems to be a sweet spot.

## 3.2 Interpretability of UCBM

**Explainable sample-wise decisions.** Figure 7 shows qualitative examples of the most contributing concepts with their contribution strength (contribution of concept $\mathbf{c}_j$ to class $y_i$: $|\mathbf{W}_{y_i,j}\pi(\mathbf{x}_i)_j|$). We find that

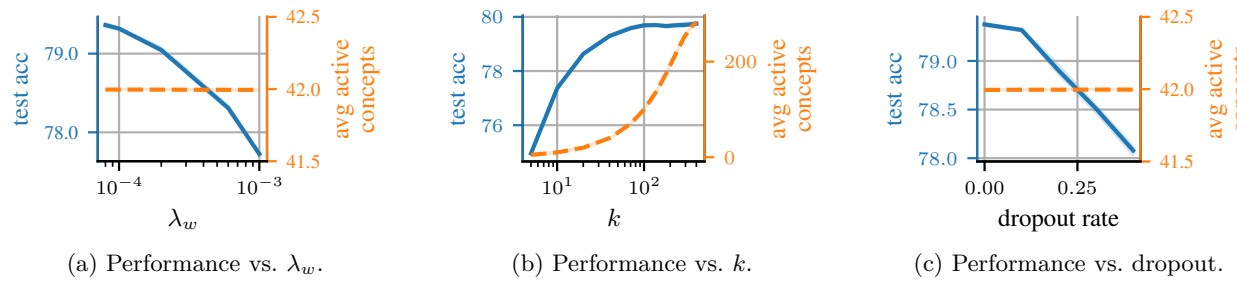

(a) Performance vs. $\lambda_w$.

(b) Performance vs. $k$.

(c) Performance vs. dropout.

Figure 6: **Sensitivity analysis over $\lambda_w$ (a), $k$ (b), and dropout (c) on ImageNet.** Larger $\lambda_w$ and smaller $k$ worsen performance, though smaller $k$ increases sparsity. There is no clear relation for dropout (also across other datasets). Results for the other datasets and concept selectors are provided in Appendix G.

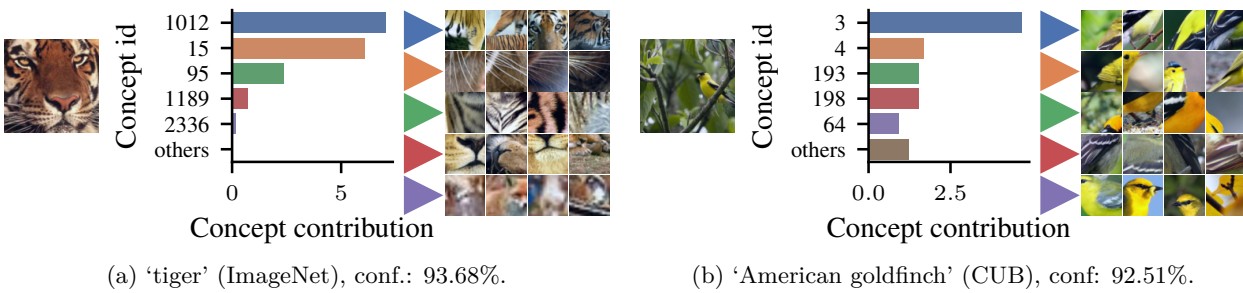

(a) 'tiger' (ImageNet), conf.: 93.68%.

(b) 'American goldfinch' (CUB), conf: 92.51%.

Figure 7: **Decisions of UCBM with TopK concept selector rely on a few reasonable and diverse concepts.** Results on ImageNet (a) and CUB (b). Additional examples are provided in Appendix H.

the most contributing concepts are relevant to both the input and prediction, while also being diverse. For example, UCBM with TopK concept selector focuses on concepts such as 'tiger striped fur', 'whiskers' or 'big cats' snouts' for the tiger in Figure 7a, or the 'bright yellow plumage' of the American goldfinch in Figure 7b.

Figure 8 compares the explanation of our UCBM with TopK concept selector, Label-free CBM, and VLG-CBM (more examples are provided in Appendix H). We find that UCBM relies on fewer concepts, that are present in the image and relevant to the predicted class. In contrast, Label-free CBM and VLG-CBM often rely on concepts that are correlated with the predicted class but absent in the image. This is especially pronounced for misclassifications (Figures 20f to 20i in Appendix H).

**User study on explainable sample-wise decisions.** To corroborate the qualitative results above, we conducted a user study to assess the interpretability of UCBM with TopK concept selector compared to Label-free CBM (we omitted VLG-CBM due to its qualitative similarity to Label-free CBM, see Figure 8 and Appendix H). Specifically, we evaluated the comprehensibility of the explanations. Note that the approaches present their concepts differently: UCBM and Label-free CBM use visual or textual concept representations, respectively. Thus, for fair comparison, we labeled concepts or retrieved images using SigLIP SoViT-400m (Zhai et al., 2023; Alabdulmohsin et al., 2023). Further details on the user study design are provided in Appendix I.

Which model is more comprehensible?

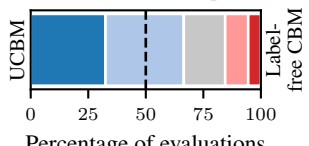

Figure 9: **Users strongly prefer UCBM.** From clearly UCBM (blue) to clearly Label-free CBM (red).

Figure 9 shows that users strongly preferred UCBM over Label-free CBM, corroborating the qualitative results shown in Figures 7 and 8 and Appendix H. Further analysis is provided in Appendix I.

**Explainable class-level decision rules.** To derive class-level decision rules, we computed the average contribution of each concept for a class. Figure 10 shows the top-3 concepts for two classes. We find that UCBM with TopK concept selector focuses on reasonable, human-understandable concepts relevant to

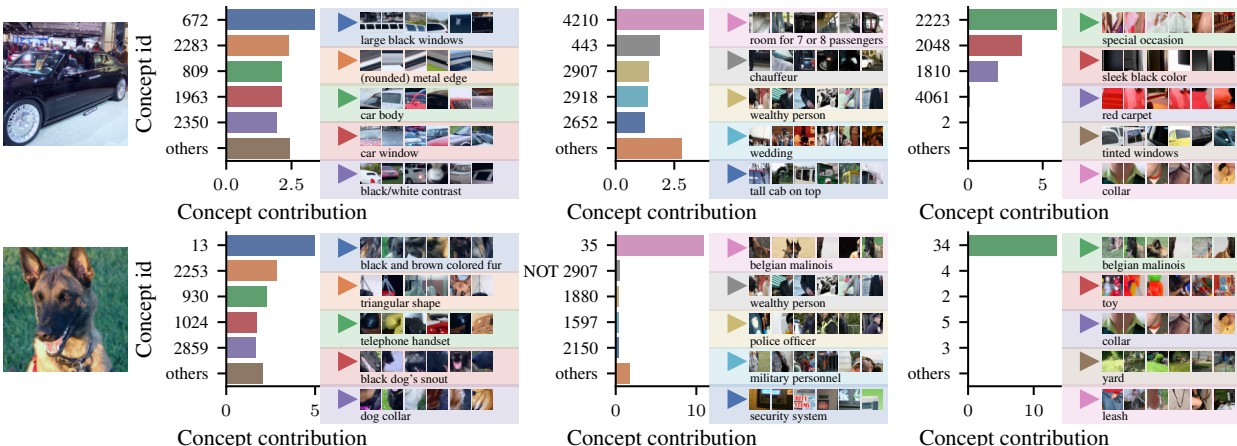

Figure 8: **The decisions of UCBM with TopK concept selector (left) are more comprehensible than those of Label-free CBM (middle) and VLG-CBM (right).** Our approach relies on concepts that are present in the image and relevant to the prediction, whereas Label-free CBM and VLG-CBM tend to use concepts that are not even present, which is particularly pronounced for misclassifications. Appendix H provides additional examples. Best viewed digitally and with zoom.

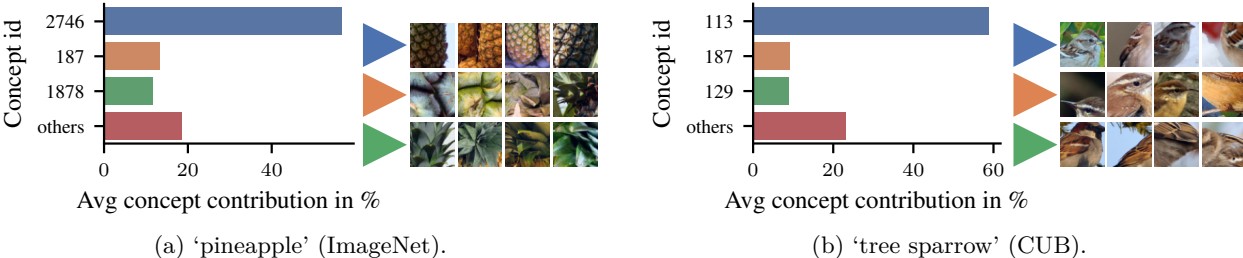

(a) 'pineapple' (ImageNet).                    (b) 'tree sparrow' (CUB).

Figure 10: **UCBM with TopK concept selector identifies class-relevant concepts** (represented by the most activating crops). Results for ImageNet (a) and CUB (b). Additional examples are provided in Appendix J.

each class. For example, Figure 10a shows that UCBM bases its classification of pineapples on the typical 'pineapple's texture' or its 'leaves'.

### 3.3 Case study: Correcting errors using a multi-modal LLM

In this subsection, we show how a multi-modal LLM (GPT-4o (Achiam et al., 2023)) can guide us to correct errors in UCBMs (specifically, a UCBM with TopK concept selector trained on ImageNet). We prompted the model asking it to adjust the weights of the sparse linear classifier $\mathbf{W}$ in UCBMs (Equation 6) to correct an error without affecting the classification of other inputs. The prompt included the misclassified input image, the top-5 concepts, and their contributions for both the misclassified and correct class. For an example of the prompt, see Appendix M. During initial experiments, we found that the suggested changes, $\Delta\mathbf{W}$, were sometimes too strong, leading to errors of previously correct inputs. To address this, we ran a grid search on the training set of ImageNet to find optimal weighing factors $\beta_i \in [0, 1]$ for each proposed change $\Delta\mathbf{W}_i$.

Figure 11 shows three examples that were correctly classified after applying the weight adjustments proposed by the LLM. This demonstrates the intervenability of UCBMs and illustrates the potential use case of multi-modal LLMs to automatically identify and correct the traceable causes of errors of UCBMs.

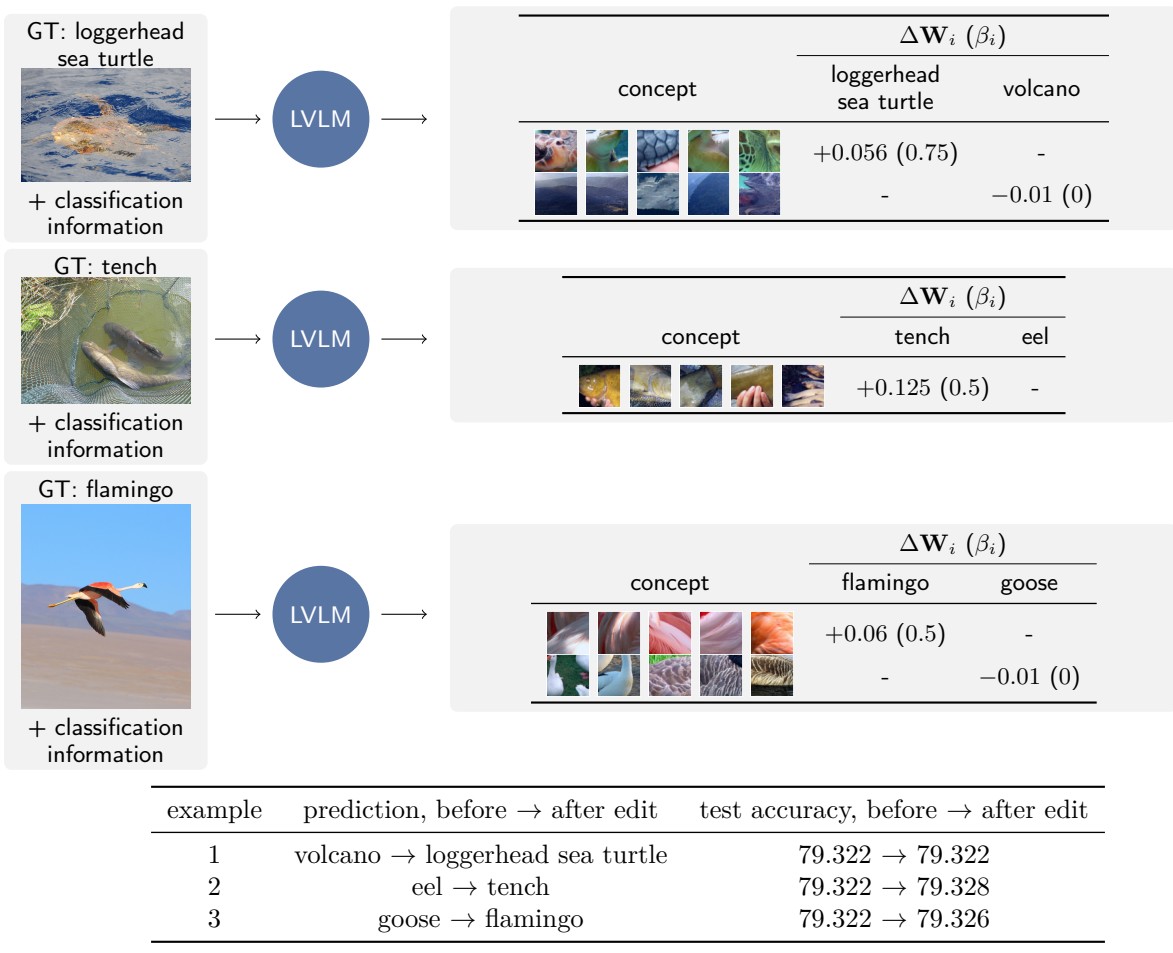

| example | prediction, before → after edit | test accuracy, before → after edit |
|---|---|---|
| 1 | volcano → loggerhead sea turtle | 79.322 → 79.322 |
| 2 | eel → tench | 79.322 → 79.328 |
| 3 | goose → flamingo | 79.322 → 79.326 |

Figure 11: **UCBMs are intervenable.** We used a multi-modal LLM to help us to correct errors by guiding the edits of the weights of UCBM with TopK concept selector ($k = 42$) that was trained on ImageNet.

## 4  Related works

**Concept-based models.**  Concept Bottleneck Models (CBMs) (Koh et al., 2020) are trained to directly leverage concepts in their classifications (Lampert et al., 2009; Kumar et al., 2009). Many works highlighted (and partially addressed) the limitations of them (Margeloiu et al., 2021; Mahinpei et al., 2021; Havasi et al., 2022; Marconato et al., 2022; Raman et al., 2024). Other work improved the performance-interpretability trade-off (Espinosa Zarlenga et al., 2022; Yang et al., 2023) or extended them beyond image classification (Ismail et al., 2023; Zarlenga et al., 2023) (see Appendix L how UCBM can be also extended to such).

The most related methods to our work convert a pretrained black-box model into a CBM post-hoc (Yuksekgonul et al., 2023; Oikarinen et al., 2023; Menon & Vondrick, 2023; Laguna et al., 2024; Dominici et al., 2024). These approaches alleviate the need for costly concept annotations by leveraging language models, like GPT-3 (Brown et al., 2020), to automatically generate class-specific descriptions and vision-language models, like CLIP (Radford et al., 2021), to learn a mapping from a black-box model's uninterpretable features to these concepts. In contrast to these, we do not presume which concepts the black-box model has learned, but find the ones that most accurately decompose the black-box model's features in an unsupervised manner. Concurrently, Rao et al. (2024) discovered concepts with sparse autoencoders to transform CLIP into a concept-based classifier. In contrast to the aforementioned works, we also introduced a novel input-dependent concept selection mechanism that dynamically retains only a sparse set of concepts per input.

**Concept discovery.** Early work searched for neuron-aligned concepts (Bau et al., 2017; Olah et al., 2017), while later works, inspired by the superposition hypothesis (Kim et al., 2018; Elhage et al., 2022), went beyond this to (linear) vector (Kim et al., 2018; Zhou et al., 2018; Olah et al., 2018; Ghorbani et al., 2019; Zhang et al., 2021; McGrath et al., 2022; Zou et al., 2023; Fel et al., 2023b; Huben et al., 2024; Stein et al., 2024), linear subspace (Vielhaben et al., 2023), or density-based (Vielhaben et al., 2024) concept representations. Early work needed costly annotated datasets to find concepts through supervision. Later work overcame this bottleneck by formulating concept discovery as a dictionary learning problem (Fel et al., 2023a).

**Model editing.** Model editing aims to modify a model's weights to remove a bias or correct errors. Previous work edited knowledge in large language models (Zhu et al., 2020; Meng et al., 2022), generative image models (Bau et al., 2020; Oldfield et al., 2023; Gandikota et al., 2023), or modified a classifier's prediction rules (Santurkar et al., 2021; Oikarinen et al., 2023). These works relied on, e.g., human intervention, factorization, or hypernetworks, whereas we leverage large vision-language models to inform model editing.

## 5 Limitations & future work

The main limitation (or advantage) of our approach is that discovered concepts are only represented visually, not textually. While images may be more informative, texts aid faster and easier interpretability. To obtain textual descriptions of concepts, we could manually label concepts. However, this does not scale to large amounts of concepts. Thus, we also experimented with automatic concept labeling through large vision-language models (GPT-4o (Achiam et al., 2023)), see Appendix K for details. While we found it to yield overall good concept descriptions, we also found many instances with poor descriptions; especially for non-object-centric or more abstract concepts. Thus, we reviewed and edited, or manually crafted concept descriptions as needed.

Another limitation of our approach is that we only extract concepts from the bottleneck layer of black-box models. We conjecture that the use of concepts throughout the feature hierarchy of these models may be beneficial for concept-based models in terms of performance and/or interpretability, as such a hierarchy is also learned by black-box models (Zeiler & Fergus, 2014). For instance, an early layer could find concepts for 'windows', 'car body', or 'wheels', while a later layer assembles them to a 'car' concept (Olah et al., 2020).

Lastly, UCBMs inherit the limitations of unsupervised concept discovery methods, such as not fully resolved polysemanticity (Graziani et al., 2024) (see Appendix B for an empirical evaluation) or concepts possibly not encoding the intended semantics (Mahinpei et al., 2021; Marconato et al., 2023; Bortolotti et al., 2024). Future work could explore discovery methods that are guaranteed (under certain conditions) to identify the true concepts encoded by a black-box model (Leemann et al., 2023).

## 6 Conclusion

We presented UCBMs, which convert pretrained black-box models into interpretable concept-based models by discovering the concepts that the model has learned through unsupervised concept discovery. We further introduced an input-dependent concept selection that effectively only retains the concepts most relevant for classifications of each input. Our experiments show that UCBMs outperform previous methods, while being substantially sparser globally. Finally, we qualitatively and quantitatively validated the interpretability of UCBMs, and showcased how multi-modal LLMs can guide the editing of UCBMs to correct its errors.

**Broader impact statement**

There are many potential positive as well as negative societal impacts of our work. However, we do not see any particular impact specific to our work that does not apply to the general impact of advancing the field of concept-based models, a subfield of machine learning.

**Acknowledgments**

This research was funded by the German Research Foundation (DFG) under grant numbers 417962828 and 499552394.

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

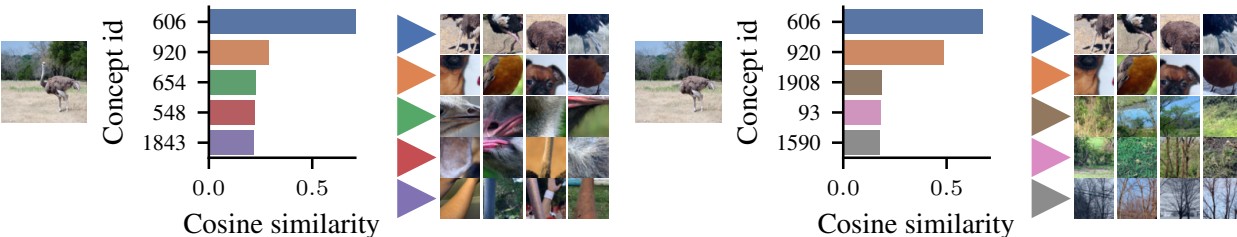

(a) Removing the head and neck of an ostrich makes concepts 654 (green), 549 (red), and 1843 (purple) disappear from the top-5 cosine similarities.

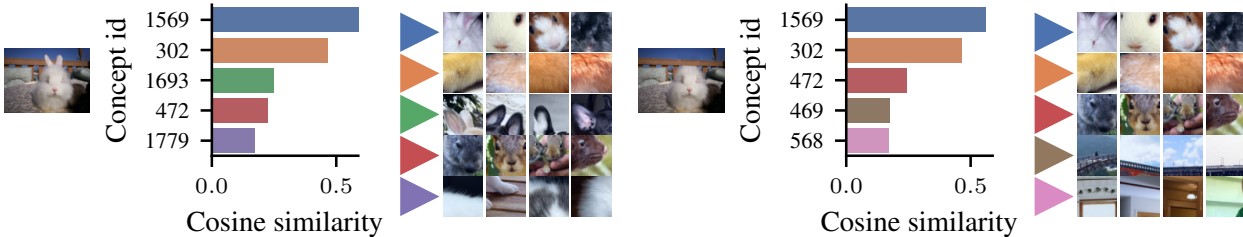

(b) Removing the ears of an angora rabbit makes concept 1693 (green) disappear from the top-5 cosine similarities.

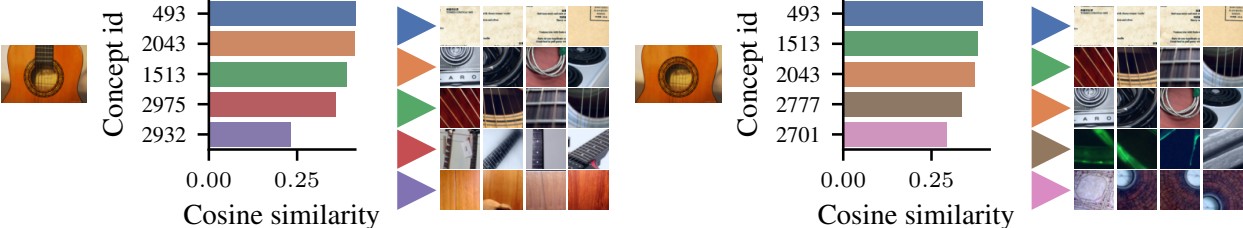

(c) Removing the neck of an acoustic guitar makes concept 2975 (red) disappear from the top-5 cosine similarities.

Figure 12: **Concepts discovered in an unsupervised manner exhibit faithful behavior.** Concepts are represented by their most activating image crops. From the original image (left), we manually removed image parts (right) and computed the concept-activation cosine similarities for an ostrich (a), angora rabbit (b), and acoustic guitar (c). We find that cosine similarity scores reduce, as we remove an image part where that concept or these concepts were previously present.

## A  Additional results for the faithfulness of discovered concepts

Figure 12 provides additional results for the faithfulness of the discovered concepts. In Figure 12a removing the head and neck of the ostrich in the input image makes concepts 654 (green), 549 (red), and 1843 (purple) disappear from the top-5 cosine similarities. Since concepts 654, 549 and 1843 represent parts of an ostrich's head or neck, this demonstrates the faithfulness of the discovered concepts. Figures 12b and 12c show similar behavior for a rabbit's ears and guitar's neck, respectively.

## B  Evaluation of concept polysemanticity

The goal of unsupervised discovery methods is to disentangle the concepts from the original feature space. That is, we typically find that individual neurons represent multiple different concepts. Although concept discovery reduces the degree of polysemanticity, recent work shows that some level of polysemanticity remains, albeit significantly reduced (Graziani et al., 2024).

To evaluate the level of polysemanticity, we visually investigated whether the top-9 most activating image crops for each concept appear to be mono- or polysemantic (see Figure 13 for examples). Specifically, we randomly selected 100 concepts discovered in our ImageNet experiment. We found that 12 of these concepts exhibited polysemanticity, while the remaining 88 appear to be monosemantic. This indicates that most

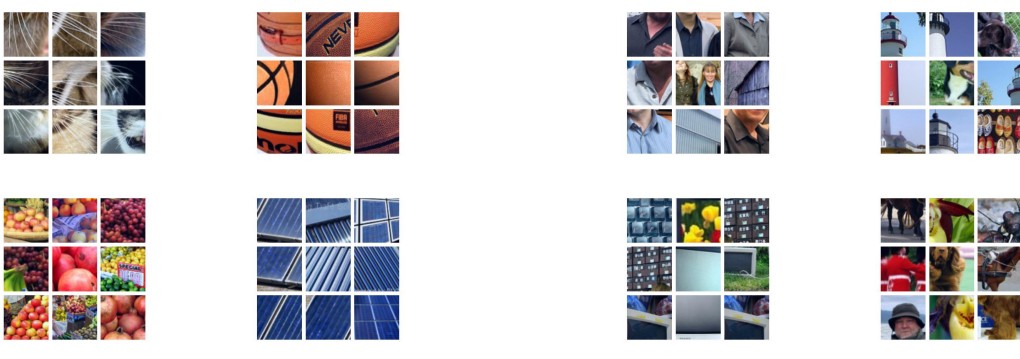

(a) Monosemantic concepts.            (b) Polysemantic concepts.

Figure 13: **Examples of mono- and polysemantic concepts.**

Table 3: **Overview of interpretable classifiers.** In the equations below, let $\tilde{s}(\mathbf{x}_i) := \text{sim}_{\mathbf{C}}(\mathbf{x}_i)$ denote the normalized cosine similarity between activations $f(\mathbf{x}_i) = \mathbf{a}_i$ for input $\mathbf{x}_i$ and the concepts $\mathbf{C}$, $\mathbf{W} \in \mathbb{R}^{|\mathcal{Y}| \times |\mathbf{C}|}$ and $\mathbf{b} \in \mathbb{R}^{|\mathcal{Y}|}$ are the weights and bias of the linear classifier, $\mathbf{o} \in \mathbb{R}_+^{|\mathbf{C}|}$ is a trainable offset parameter, $y_i \in \mathcal{Y}$ denotes the target class of input $\mathbf{x}_i$ for a total of $|\mathcal{Y}|$ classes, $\mathcal{L}$ denotes the task-specific loss function (cross-entropy loss throughout this work), $R_\alpha$ is the elastic net regularization penalty (Zou & Hastie, 2005), $\lambda_w$, $\lambda_\pi$ govern the regularization strengths, $H$ denotes the Heaviside step function, and TopK denotes the TopK activation function (Makhzani & Frey, 2014). Note that $\tilde{s}(\mathbf{x}_i)$ is frozen during optimization. Further, note that the TopK concept selector does not need a sparsity penalty since sparsity can be controlled directly using the hyperparameter $k$.

| name | concept selector $\pi$ | interpretable classifier |
|---|---|---|
| ReLU | $\pi(\mathbf{x}_i) := \max(0, \tilde{s}(\mathbf{x}_i) - \mathbf{o})$ | $\min_{\mathbf{W},\mathbf{b},\mathbf{o}} \sum_{i=1}^{N} \mathcal{L}(\mathbf{W}\pi(\mathbf{x}_i) + \mathbf{b}, y_i) + \lambda_w R_\alpha(\mathbf{W}) + \lambda_\pi R_\alpha(\pi(\mathbf{x}_i))$ |
| JumpReLU | $\pi(\mathbf{x}_i) := \tilde{s}(\mathbf{x}_i) \cdot H(\tilde{s}(\mathbf{x}_i) - \mathbf{o})$ | $\min_{\mathbf{W},\mathbf{b},\mathbf{o}} \sum_{i=1}^{N} \mathcal{L}(\mathbf{W}\pi(\mathbf{x}_i) + \mathbf{b}, y_i) + \lambda_w R_\alpha(\mathbf{W}) + \lambda_\pi \sum_{j}^{|\mathbf{C}|} H(\tilde{s}_j(\mathbf{x}_i) - \mathbf{o}_j)$ |
| TopK | $\pi(\mathbf{x}_i) := \text{TopK}_k(\tilde{s}(\mathbf{x}_i) - \mathbf{o})$ | $\min_{\mathbf{W},\mathbf{b},\mathbf{o}} \sum_{i=1}^{N} \mathcal{L}(\mathbf{W}\pi(\mathbf{x}_i) + \mathbf{b}, y_i) + \lambda_w R_\alpha(\mathbf{W})$ |

concepts discovered by the unsupervised discovery are monosemantic, but a number of concepts still retain polysemantic characteristics.

## C  Further details on the interpretable classifiers

Table 3 provides the complete overview of the interpretable classifiers for all of our UCBM variants from Section 2.2. Below, we provide further details for the JumpReLU and TopK concept selectors.

**JumpReLU concept selector.** The JumpReLU activation function (Erichson et al., 2019) is defined as follows:

$$\text{JumpReLU}_{\mathbf{o}}(\mathbf{x}) = \mathbf{x} \cdot H(\mathbf{x} - \mathbf{o}) = \begin{cases} 0, & \mathbf{x} \le \mathbf{o} \\ \mathbf{x}, & \mathbf{x} > \mathbf{o} \end{cases}, \tag{8}$$

where $H$ is the Heaviside step function. Note that we cannot directly train our offset parameter $\mathbf{o}$. Thus, following Rajamanoharan et al. (2024), we used straight-through-estimators (Bengio et al., 2013) to make $\mathbf{o}$ trainable. Specifically, we adopted the pseudo-derivates from Rajamanoharan et al. (2024):

$$\frac{\tilde{\partial}}{\partial \mathbf{o}} \text{JumpReLU}_{\mathbf{o}}(\mathbf{x}) := -\frac{\mathbf{o}}{\epsilon} K(\frac{\mathbf{x} - \mathbf{o}}{\epsilon}) \tag{9}$$

Table 4: **Hyperparameter settings for all UCBMs variants on ImageNet | CUB | Places-365.**

|  | $\lambda_\pi$ | $k$ | $\lambda_w$ | dropout rate |
|---|---|---|---|---|
| UCBM w/o concept selection | n/a | n/a |  |  |
| UCBM with ReLU concept selector | 2e-5 \| 1e-4 \| 2e-5 | n/a |  |  |
| UCBM with JumpReLU concept selector | 1e-5 \| 4e-7 \| 4e-7 | n/a | 1e-4 \| 8e-4 \| 4e-4 | 0.1 \| 0.2 \| 0.2 |
| UCBM with TopK concept selector | n/a | 42 \| 66 \| 162 |  |  |

and

$$\frac{\tilde{\partial}}{\tilde{\partial}\mathbf{o}}H(\mathbf{x}-\mathbf{o}) := -\frac{1}{\epsilon}K\left(\frac{\mathbf{x}-\mathbf{o}}{\epsilon}\right) \quad, \tag{10}$$

where $\tilde{\partial}$ denotes the pseudo-derivative, $K$ is a kernel (following Rajamanoharan et al. (2024) we used the rectangle function: $\text{rect}(\mathbf{x}) := H(\mathbf{x}+\frac{1}{2}) - H(\mathbf{x}-\frac{1}{2})$), and $\epsilon$ can be seen as the KDE bandwidth.

**TopK concept selector.** The TopK activation function (Makhzani & Frey, 2014) is defined as follows:

$$\text{TopK}_k(\mathbf{x})_i = \begin{cases} \mathbf{x}_i & \text{if } \mathbf{x}_i \in \text{top-k}(\mathbf{x}), \\ 0 & \text{otherwise,} \end{cases} \quad . \tag{11}$$

We used the TopK implementation of Gao et al. (2025) who internally apply a non-linearity (we used ReLU) after the actual TopK function. Note that we can directly control the sparsity through the hyperparameter $k$ and the TopK concept selector becomes equivalent to the identity function as $k = |\mathbf{C}|$.

**Why do we add a trainable offset parameter o?** We introduce the additional trainable offset parameter $\mathbf{o} \in \mathbb{R}_+^{|\mathbf{C}|}$ to allow the classifier to adapt to different ranges of alignment scores for each concept. The reasons for this is that the distribution of scores can vary between concepts. For example, for one concept, the scores may be more uniformly distributed, indicating a more ambiguous presence of the concept. For another concept, the scores might follow a bimodal distribution, indicating two distinct modes that indicate the object is present or absent. The offset parameter helps the classifier in such cases to account for such different distributions.

## D  Hyperparameter settings

Table 4 provides the hyperparameters ($\lambda_\pi$, $k$, $\lambda_w$, dropout rate) for all our UCBMs variants. We chose those hyperparameters such that they yielded a good trade-off between performance, sparsity, and fair comparability (see Figure 6 and Appendices E and G). It is important to note that we first optimized $\lambda_\pi$ for the ReLU and JumpReLU concept selectors and then set $k$ accordingly, as we found that its relationship to sparsity (c.f., Equation 7) is straightforward.

## E  Number of concepts used per class prediction

Table 5 provides the absolute numbers for Table 1. UCBM variants with concept selector use substantially fewer concepts than prior CBMs and UCBM without concept selection or binary indicator (Panousis et al., 2023).

Beyond the sparsity measurements in Tables 1 and 5, we computed how many concepts the models need to explain their prediction of a class. For this, we computed the mean number of concepts that are required to explain 95% (or 90%) of a model's prediction per sample:

$$\frac{1}{N}\sum_{i=1}^{N} C_i' \text{ , where } \min_{C_i' \subseteq \{1,\ldots,|\mathbf{C}|\}} |C_i'| \text{ s.t. } \frac{\sum\limits_{c \in C_i'} |\mathbf{W}_{\tilde{y}_i,c}\pi(\mathbf{x}_i)_c|}{\sum\limits_{c \in \{1,\ldots,|\mathbf{C}|\}} |\mathbf{W}_{\tilde{y}_i,c}\pi(\mathbf{x}_i)_c|} \geq 95\% \quad , \tag{12}$$

Table 5: **The concept selection mechanism leads to substantially fewer concepts being used in the classification.** We report the mean number of active concepts with standard deviation according to Equation 7. Parentheses show the total number of concepts $|\mathbf{C}|$. Label-free CBM, VLG-CBM, UCBM without concept selection, and UCBM with binary indicator use many more concepts than our UCBM variants with concept selection.

| | Mean number of active concepts (c.f. Equation 7) | | |
|---|---|---|---|
| Method | ImageNet | CUB | Places-365 |
| Post-hoc CBM (Yuksekgonul et al., 2023) | n/a | 112.0 (112) | n/a |
| Label-free CBM (Oikarinen et al., 2023) | 4238.0 (4521) | 211.9 (212) | 1820.0 (2008) |
| VLG-CBM (Srivastava et al., 2024) | 3018.97 (4300) | 661.99 (671) | 1382.99 (2186) |
| UCBM w/o concept selection | 3000.0 (3000) | 200.0 (200) | 1825.0 (1825) |
| UCBM with binary indicator (Panousis et al., 2023) | 1995.7 (3000) | 200.0 (200) | 899.3 (1825) |
| UCBM with ReLU concept selector | 47.8 (3000) | **61.0** (200) | *162.4* (1825) |
| UCBM with JumpReLU concept selector | *42.8* (3000) | *62.3* (200) | 166.2 (1825) |
| UCBM with TopK concept selector | **42.0** (3000) | 64.2 (200) | **162.0** (1825) |

Table 6: **UCBM with TopK concept selector requires less concepts to explain a prediction**. We report the mean and the standard deviation of the number of concepts that are required to explain 95% of the prediction (see Equation 12 for more details).

| | #concepts to explain 95% of the prediction (Equation 12) | | |
|---|---|---|---|
| Approach | ImageNet | CUB | Places-365 |
| UCBM w/o concept selection | $8.79 \pm 8.093$ | $5.79 \pm 1.774$ | $46.1 \pm 11.594$ |
| UCBM with ReLU concept selector | $3.83 \pm 2.323$ | $4.7 \pm 1.586$ | $15.72 \pm 4.032$ |
| UCBM with JumpReLU concept selector | $5.05 \pm 3.334$ | $4.53 \pm 1.679$ | $25.05 \pm 8.068$ |
| UCBM with TopK concept selector | $4.95 \pm 2.933$ | $5.25 \pm 1.747$ | $24.72 \pm 8.04$ |

where $\tilde{y}_i$ denotes the model's prediction of input $\mathbf{x}_i$.

Table 6 shows that UCBMs with concept selector rely on fewer concepts than UCBM without concept selection. Note that relying on fewer concepts makes it easier for users to comprehend a prediction since they do not need to inspect a lot of concepts.

Figure 14 provides a detailed per-class analysis of the mean number of concepts required to predict each class, i.e., to explain 90% or 95% of the prediction (Equation 12). While some classes only rely on a few or even only a single concept, others require substantially more. For example, on ImageNet, certain classes—such as 'goldfish', 'great grey owl', 'trilobite', 'quail', 'hornbill', 'abacus', 'bell or wind chime', 'harp', 'jigsaw puzzle', 'marimba', 'maze', 'graduation cap', 'mousetrap', 'piggy bank', 'pinwheel', 'pool table', 'solar thermal collector', 'umbrella', 'water tower', 'crossword', 'jackfruit', and 'horse chestnut seed'—typically only use a single concept. In contrast, other classes—such as 'Redbone Coonhound', 'Tibetan Terrier', 'Golden Retriever', 'patas monkey', 'titi monkey', and 'monastery'—require substantially more concepts on average (13).

## F  Other feature encoder choices

Table 7 shows the results for InceptionV3 (Szegedy et al., 2015) and ViT-B/16 (Dosovitskiy et al., 2021) feature encoders, both pre-trained on ImageNet.[4] Consistent with the findings in Section 3.1, UCBM with TopK concept selector achieves performance close to the original, black-box models. While we maintained

---

[4]Both (black-box) models are provided at https://github.com/pytorch/vision.

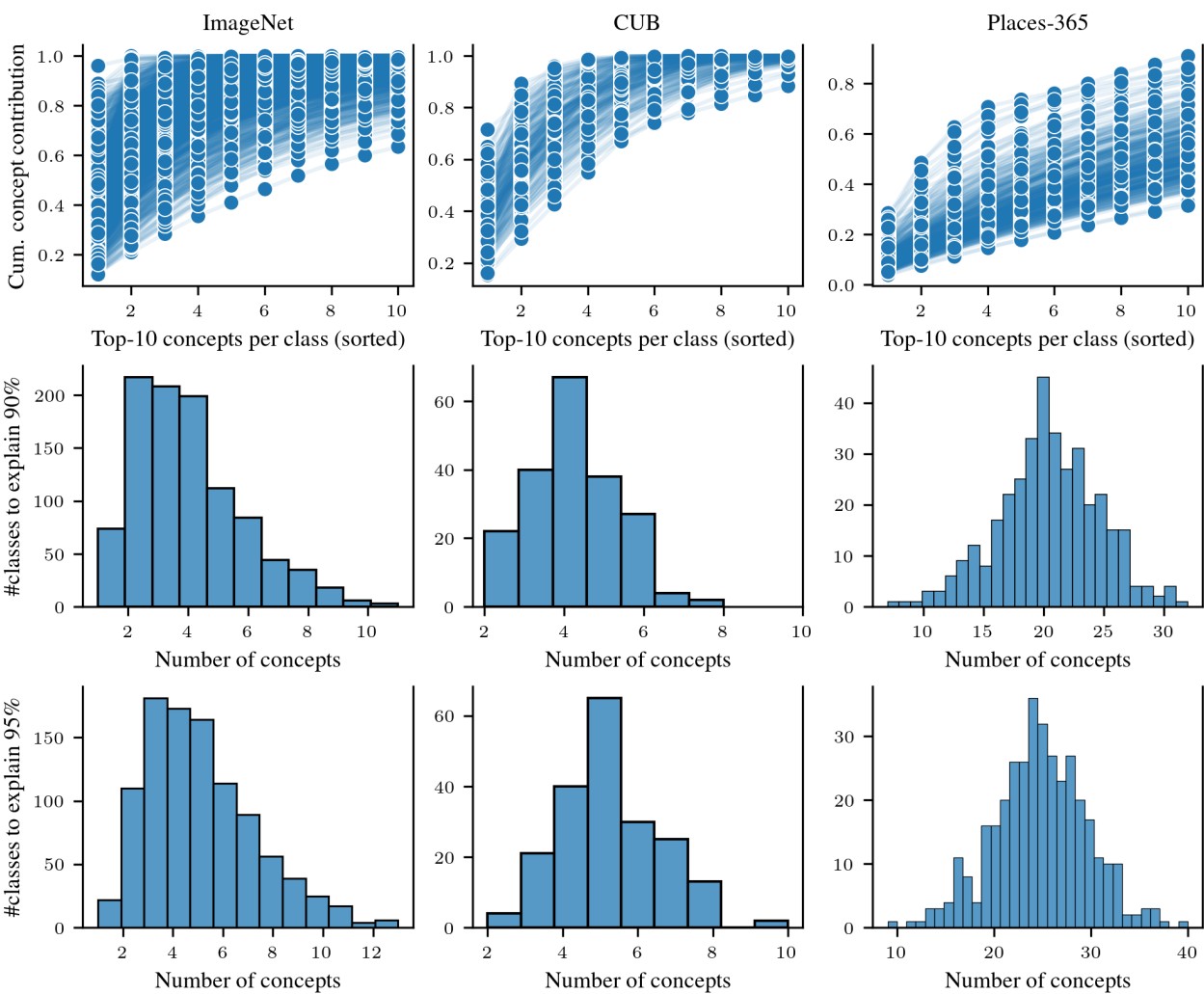

Figure 14: **Analysis of number of concepts used per class.** Top panel: Cumulative contribution of the top 10 concepts per class. Middle and bottom panels: Number of concepts required to reach 90% or 95% cumulative contribution per class, respectively. While some classes rely on only a few concepts, others rely on more.

Table 7: **UCBM also performs well with InceptionV3 and ViT backbone** on ImageNet.

| Method | ImageNet top-1 test accuracy |
|---|---|
| Original InceptionV3 | 77.29 |
| UCBM w/ TopK | 73.23 |
| Original ViT-B/16 | 81.01 |
| UCBM w/ TopK | 77.94 |

the same sparsity level for UCBM when using InceptionV3 as the feature encoder, applying the same level to ViT led to a performance drop, necessitating a reduced level of sparsity. We hypothesize that non-negative matrix factorization may not be the most effective approach for extracting concepts from ViT's *non-negative* feature space. Exploring alternative concept discovery methods, such as sparse autoencoders, could allow us to restore higher levels of sparsity with less performance compromises.

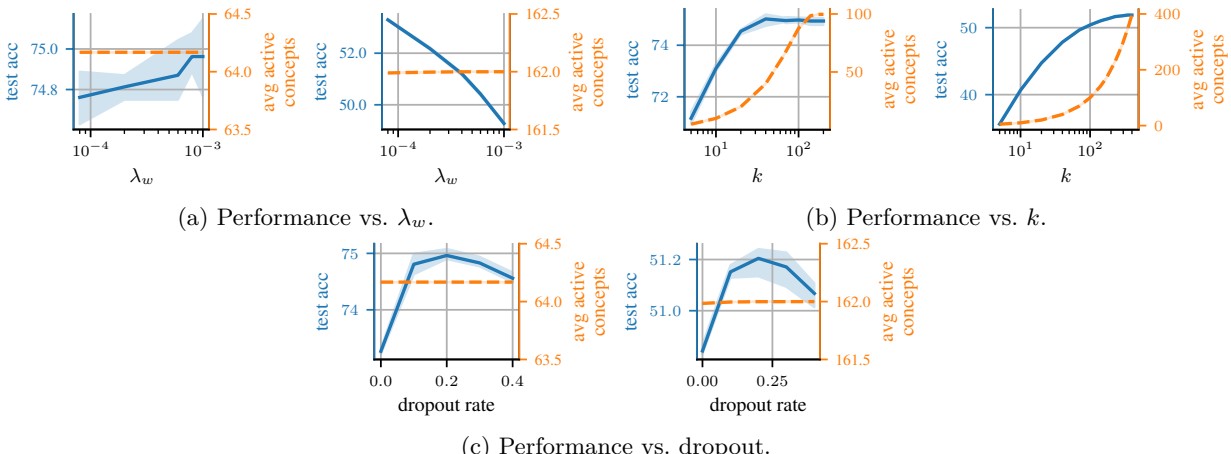

(a) Performance vs. $\lambda_w$.

(b) Performance vs. $k$.

(c) Performance vs. dropout.

Figure 15: **Sensitivity analysis for UCBM with TopK concept selector over $\lambda_w$ (a), $k$ (b), and the dropout rate (c) for CUB (left) and Places-365 (right).**

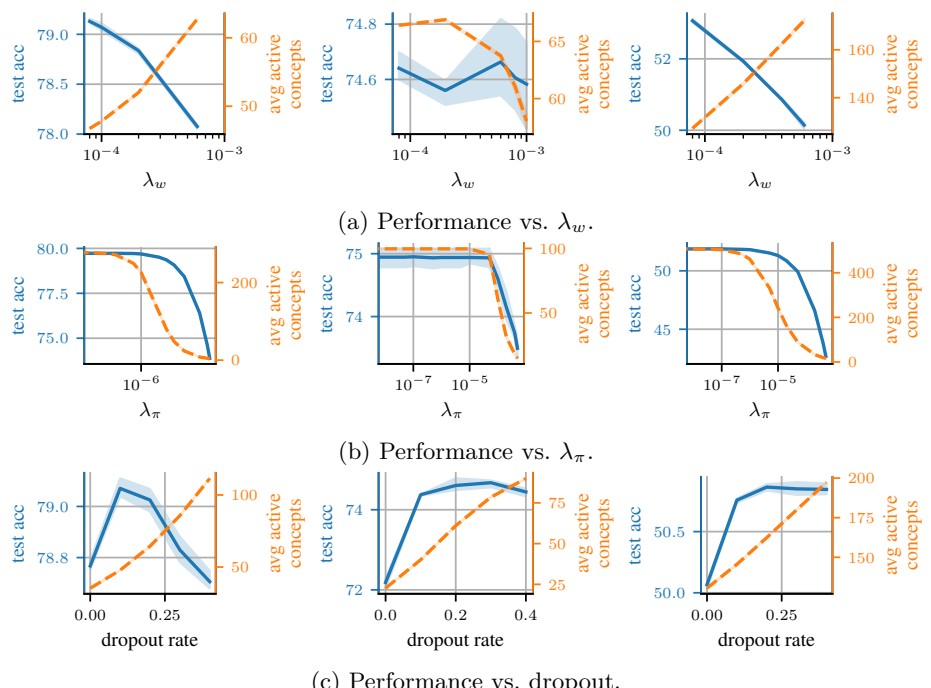

(a) Performance vs. $\lambda_w$.

(b) Performance vs. $\lambda_\pi$.

(c) Performance vs. dropout.

Figure 16: **Sensitivity analysis for UCBM with ReLU concept selector over $\lambda_w$ (a), $\lambda_\pi$ (b), and the dropout rate (c) for ImageNet (left), CUB (middle), and Places-365 (right).**

## G   Additional sensitivity analysis results

Figure 15 provides the results for the sensitivity analysis for UCBM with TopK concept selector on CUB and Places-365. Figures 16 and 17 provide the results for UCBM with ReLU or JumpReLU concept selector, respectively.

We find that the hyperparameters $k$ (for TopK) or $\lambda_\pi$ (for ReLU and JumpReLU) control the trade-off between performance and sparsity (see also Figure 3). Regarding the other hyperparameters, $\lambda_w$ and dropout rate, it is important to observe that they have less influence on the sparsity for the TopK concept selector than for the other concept selectors. We consider this as an advantage of the TopK concept selector, as it

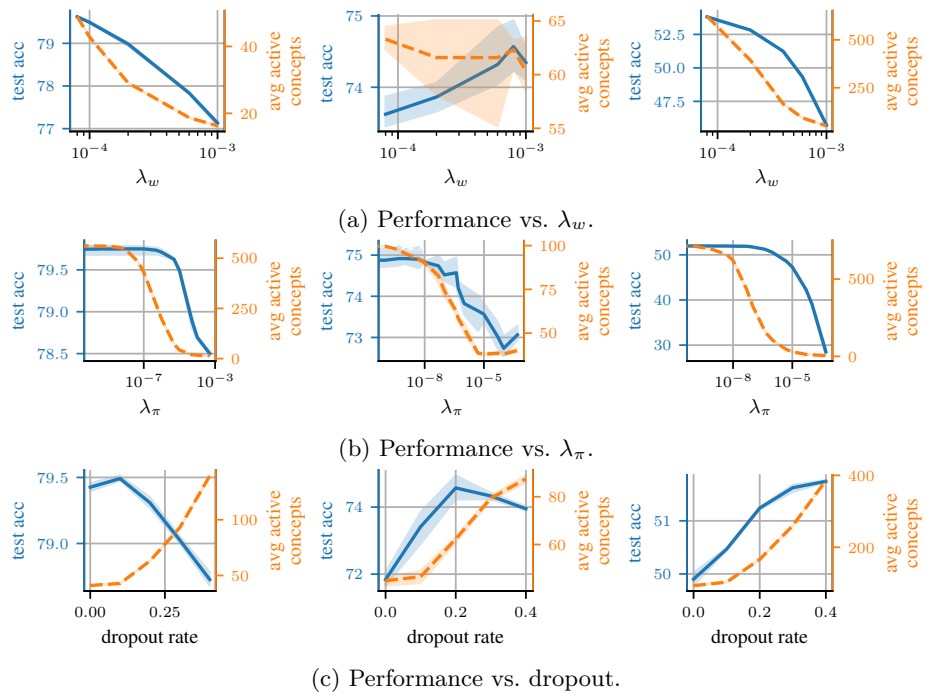

(a) Performance vs. $\lambda_w$.

(b) Performance vs. $\lambda_\pi$.

(c) Performance vs. dropout.

Figure 17: **Sensitivity analysis for UCBM with JumpReLU concept selector over $\lambda_w$ (a), $\lambda_\pi$ (b), and the dropout rate (c) for ImageNet (left), CUB (middle), and Places-365 (right).**

reduces the interaction between hyperparameters. This makes hyperparameter tuning simpler and simplifies the interpretation: $k$ governs the average number of active concepts per sample, $\lambda_w$ governs the number of concepts used per class, and the dropout rate influences whether the classifier relies on a broader or narrower set of concepts.

For $\lambda_w$, we find that increasing it typically leads to worse performance and a smaller average number of active concepts per sample. Interestingly, for the UCBMs with ReLU concept selector trained on ImageNet and Places-365, we observe the opposite behavior. For the dropout rate, a higher dropout rate results in more active concepts per sample, though its relationship with performance is less clear.

## H   Additional examples of explainable decisions

**Additional examples for sample-wise explanations.**   Figure 18 provides more examples of explainable decision of UCBM with TopK concept selector on ImageNet, CUB, and Places-365. We typically find that our method relies on a small set of concepts that are present in the images, human-comprehensible and class-relevant. For instance, for the viaduct in Figure 18a, UCBM uses class-relevant concepts (e.g., 'arches', 'stones', or 'walkway'). For the 'railroad track' in Figure 18c, it uses concepts such as 'tracks' or 'train'. Interestingly, it also uses the concept 'large window' that is also related to, e.g., buses. This indicates that UCBMs first assess if concepts are present or absent and then based on that evidence predict the class that is most likely given that.

**Understanding misclassifications of UCBMs.**   Figure 19 shows that we can comprehend why UCBMs made a misclassification. For example, Figure 19a shows that the UCBM incorrectly predicted 'car wheel' instead of 'station wagon'. However, the image shows such station wagon mirrored in a car wheel. Looking at the most contributing concepts reveals that UCBM focused on concepts that are related to the car wheel, as it is the most salient in the image.

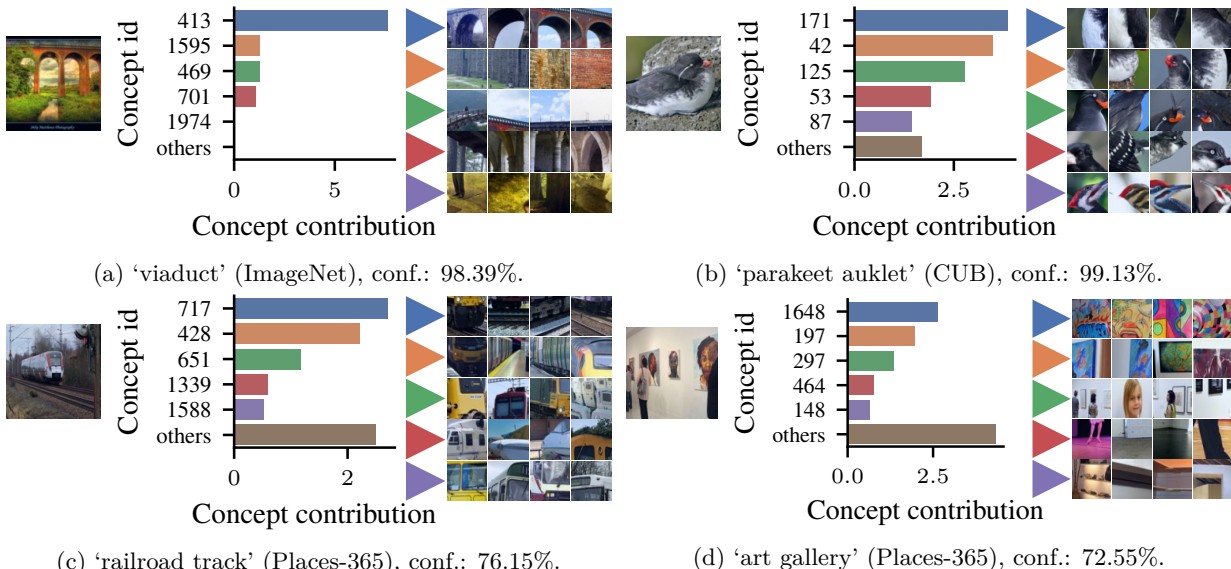

(a) 'viaduct' (ImageNet), conf.: 98.39%.

(b) 'parakeet auklet' (CUB), conf.: 99.13%.

(c) 'railroad track' (Places-365), conf.: 76.15%.

(d) 'art gallery' (Places-365), conf.: 72.55%.

Figure 18: **Explainable decisions by UCBM** with TopK concept selector on ImageNet (a), CUB (b), and Places-365 (c, d) classes. The model's prediction are comprehensible and typically rely on only few concepts.

**Additional examples for the comparison of UCBM to Label-free CBM and VLG-CBM.** Figure 20 compares the explanations of UCBM with TopK concept selector, Label-free CBM (Oikarinen et al., 2023), and VLG-CBM (Srivastava et al., 2024). We find that our approach provides more comprehensible explanations:[5] UCBM relies on intuitive concepts that are present in the image and relevant to the prediction. In contrast, Label-free CBM and VLG-CBM tend to rely on concepts that are correlated to the predicted class but may not be present in the image, e.g., the concepts 'graduation markings' or 'graduation ceremony' for the prediction 'graduation cap' in Figure 20d.[6] We quantified the usage of non-visible concepts in the predictions of each method in Table 8. Note that such reliance on prediction-class correlated but absent concepts is particularly pronounced for misclassifications (Figures 20f to 20i and Table 8). For example in Figure 20h showing a broom nearby a lake, Label-free CBM relies on the concepts 'mellow, flute-like sound', 'wind instrument', or 'bagpipe'. Similarly, VLG-CBM relies on the concepts 'tangled twisted shape', 'made of rope or string', or 'Mexican food'. None of these concepts are present in the image. We believe relying on such non-visible concepts is not helpful to understand the decision of a concept-based model.

# I Further details on the user study

In the user study, we studied whether users consider the explanations of the decisions of UCBM to be comprehensible. To do so, we compared the explanations of UCBM with TopK concept selector with Label-free CBM (Oikarinen et al., 2023). Both were trained on ImageNet.

**Task.** We asked users to assess which model provides a more comprehensible explanation from a scale from 'Model A clearly more' to 'Model B clearly more'. Further, we asked for the reasons why they think one model is more comprehensible than the other.

---

[5]These qualitative findings are further corroborated in the user study in Section 3.2 and Appendix I.

[6]We suspect the reason for this are shortcomings of the vision language models used in both approaches. For instance, the concept '*graduated* cylinder' is unrelated to the prediction of 'graduation cap' in Figure 20d. However, the word 'graduated' is related to 'graduation'. Indeed, when we compute the cosine similarity of text features (we considered the following: 'graduated cylinder', 'graduation ceremony', 'graduation markings', 'graduation', 'university', 'dog', 'house'), we found that concepts related graduation have higher similarities with the graduated cylinder than the unrelated concepts. We leave further investigations for future work.

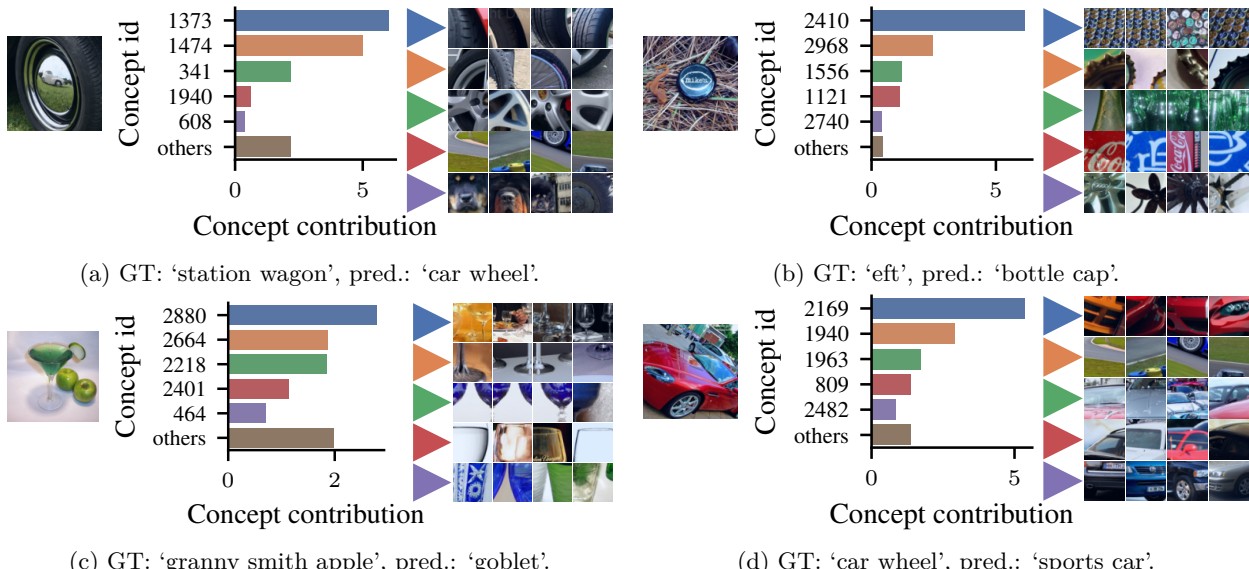

(a) GT: 'station wagon', pred.: 'car wheel'.

(b) GT: 'eft', pred.: 'bottle cap'.

(c) GT: 'granny smith apple', pred.: 'goblet'.

(d) GT: 'car wheel', pred.: 'sports car'.

Figure 19: **The most contributing concepts explain the misclassifications on ImageNet of UCBM with TopK concept selector. a**: The image shows a station wagon mirrored in a car wheel. Most of the top-5 concepts are related to car wheels, which explains that the model only focuses on the car wheel itself instead of the mirrored station wagon. This clearly explains why the model predicts 'car wheel' instead of 'station wagon'. **b**: The image shows an eft next to a bottle cap. The concepts show that the model used concepts related to bottle caps, which is the object at the center of the image. **c**: The image shows two granny smith apples next to a goblet that was predicted by the model. The concepts reveal that the model focuses on concepts related to the goblet at the center of the image. **d**: The image shows a sports car, including one of its front wheels. The most important concept is related to sports cars. The other concepts also focus more on general car concepts than on the wheels.

**User study data.** We showed users sample-wise (local) explanations based on which concepts contributed the most to the decision of each model, akin to Figures 8, 10, 18 and 19. Importantly, 20% of samples showed misclassifications of *both* models (for the other 80% both model predicted correctly).[7] We include misclassifications to also understand how comprehensible models are under errors. We believe this is an important aspect to study, as users will also interact with models that make errors in practice. For sake of this user study, we simplified the explanations by removing the concept contributions and only showed the names and top-activating image crops of the five most contributing concepts and a corresponding concept description.

Note that UCBM and Label-free CBM represent their concepts differently: UCBMs show visual representations, whereas Label-free CBM shows concept descriptions. To ensure fair comparison, we labeled the most activating image crops of UCBM's concepts and retrieved images using SigLIP SoViT-400m (Zhai et al., 2023; Alabdulmohsin et al., 2023) for Label-free CBM's concepts.

**Setup.** We implemented the user study in a lightweight Python GUI so that users could run the study locally on their machines. Users were provided with the task description (Figure 21) and an example (Figure 22). After the instruction, users interacted with our user study interface (Figure 23).

We asked ten users to rate a total of 200 samples (20 per user). Users participated voluntarily and without payment. They have strong background in machine learning and related fields. However, none of them is working on concept-based models or had seen explanations of UCBM before.

---

[7]No sample for which one model was correct and the other was incorrect was shown in the user study.

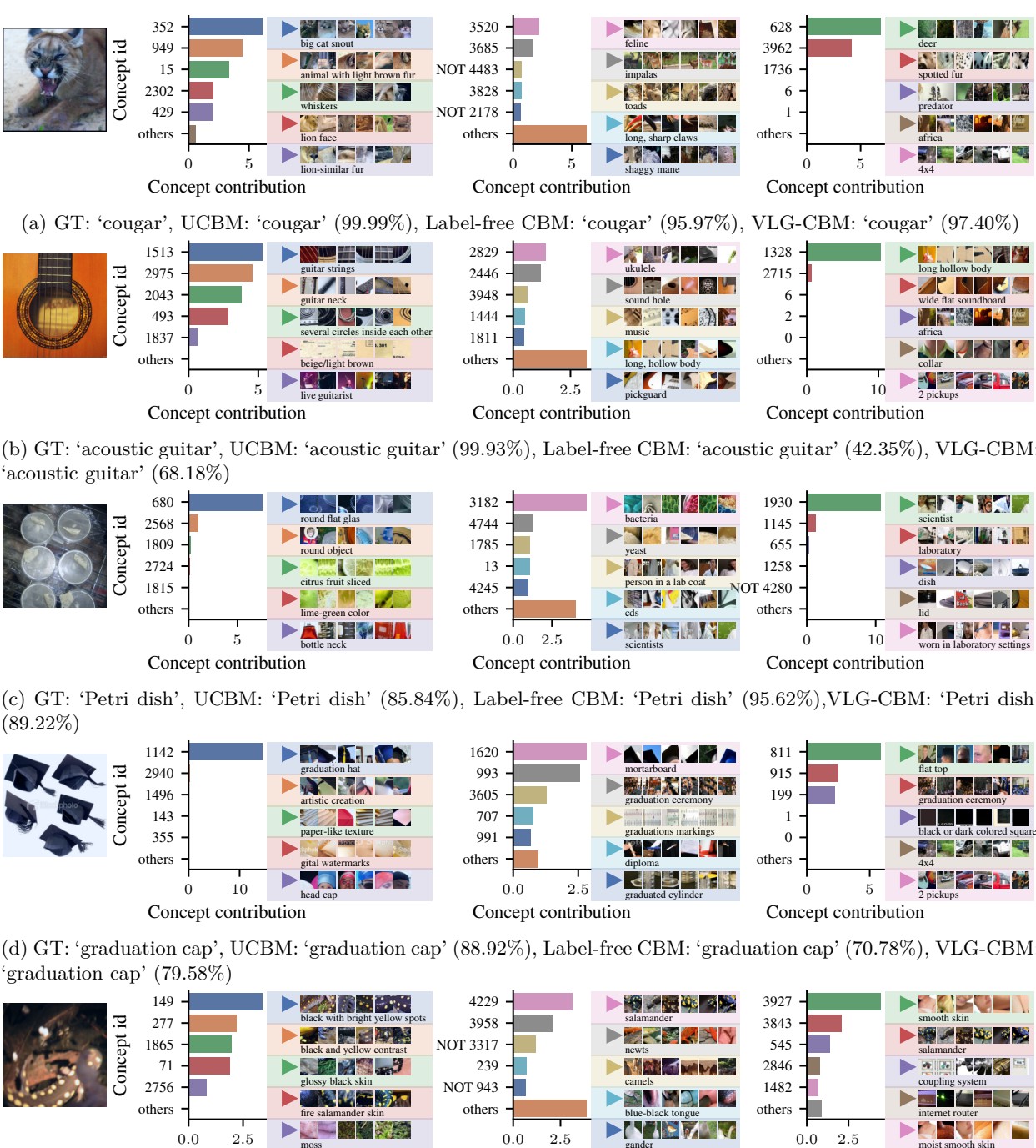

(a) GT: 'cougar', UCBM: 'cougar' (99.99%), Label-free CBM: 'cougar' (95.97%), VLG-CBM: 'cougar' (97.40%)

(b) GT: 'acoustic guitar', UCBM: 'acoustic guitar' (99.93%), Label-free CBM: 'acoustic guitar' (42.35%), VLG-CBM: 'acoustic guitar' (68.18%)

(c) GT: 'Petri dish', UCBM: 'Petri dish' (85.84%), Label-free CBM: 'Petri dish' (95.62%), VLG-CBM: 'Petri dish' (89.22%)

(d) GT: 'graduation cap', UCBM: 'graduation cap' (88.92%), Label-free CBM: 'graduation cap' (70.78%), VLG-CBM: 'graduation cap' (79.58%)

(e) GT: 'spotted salamander', UCBM: 'spotted salamander' (90.35%), Label-free CBM: 'spotted salamander' (92.77%), VLG-CBM: 'spotted salamander' (77.39%)

**Further analysis.** Complementary to the results presented in Section 3.2, we conducted further analysis on the results of the user study. Figure 24 shows that users strongly preferred our UCBM with Topk concept selector over Label-free CBM in ca. 65-70% of evaluations (Label-free CBMs are only preferred in ca. 15%). Users' preference was similar for correct or incorrect predictions.

Users based their preference decisions mostly on relevance to the prediction (selected in 66.5% of the evaluations). However, relevance to the image (55%) and informativeness (55%) closely followed it.

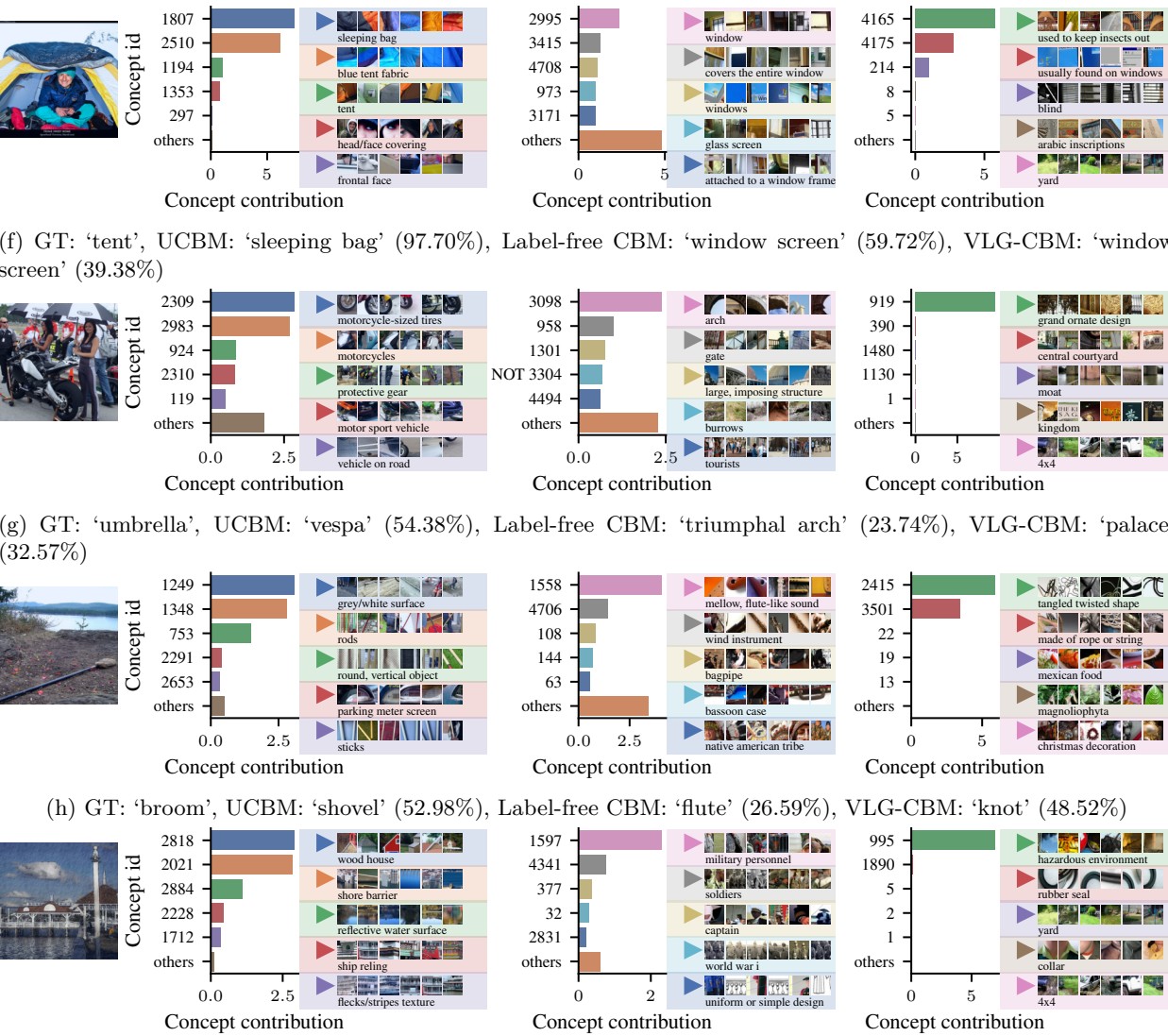

(f) GT: 'tent', UCBM: 'sleeping bag' (97.70%), Label-free CBM: 'window screen' (59.72%), VLG-CBM: 'window screen' (39.38%)

(g) GT: 'umbrella', UCBM: 'vespa' (54.38%), Label-free CBM: 'triumphal arch' (23.74%), VLG-CBM: 'palace' (32.57%)

(h) GT: 'broom', UCBM: 'shovel' (52.98%), Label-free CBM: 'flute' (26.59%), VLG-CBM: 'knot' (48.52%)

(i) GT: 'pier', UCBM: 'boathouse' (32.64%), Label-free CBM: 'military uniform' (6.46%), VLG-CBM: 'gas mask' (19.59%)

Figure 20: **Comparison of explainable decisions of UCBM with TopK concept selector (left) vs. Label-Free CBM (middle) vs. VLG-CBM (right)**. Subfigures a-e and f-i show correct or incorrect predictions of the CBMs, respectively. Our UCBM with TopK concept selector provides more comprehensible explanations, while Label-free CBM and VLG-CBM often rely on concepts that are not even visible in the image (this is especially pronounced for misclassifications).

## J    Additional examples of explainable decision rules

Figure 25 provides more examples of explainable decision rules of UCBM. The examples show that UCBM uses reasonable human-interpretable concepts to build the score of a specific class.

Table 8: **Number of concepts used in the predictions that are actually visible in the image.** We report the numbers concepts that are actually visible in the image by inspecting Figures 8 and 20. Our UCBM reliably uses concepts that are actually visible within in the image. In contrast, Label-free CBM and VLG-CBM frequently use concepts that, while relevant to the predicted class, are not actually present in the given image.

Correct predictions (Figures 8 and 20a to 20e; total of 7)

| $i$-th most important concept | UCBM | Label-free CBM | VLG-CBM |
|---|---|---|---|
| 1st | 7/7 | 5/7 | 3/7 |
| 2nd | 6/7 | 3/7 | 4/7 |
| 3rd | 6/7 | 0/7 | 4/7 |
| 4th | 7/7 | 2/7 | 3/7 |
| 5th | 5/7 | 2/7 | 1/7 |

Incorrect predictions (Figures 20f to 20i; total of 4)

| $i$-th most important concept | UCBM | Label-free CBM | VLG-CBM |
|---|---|---|---|
| 1st | 4/4 | 1/4 | 1/4 |
| 2nd | 3/4 | 0/4 | 0/4 |
| 3rd | 4/4 | 0/4 | 0/4 |
| 4th | 3/4 | 0/4 | 0/4 |
| 5th | 4/4 | 1/4 | 0/4 |

Figure 21: **Instruction text.**

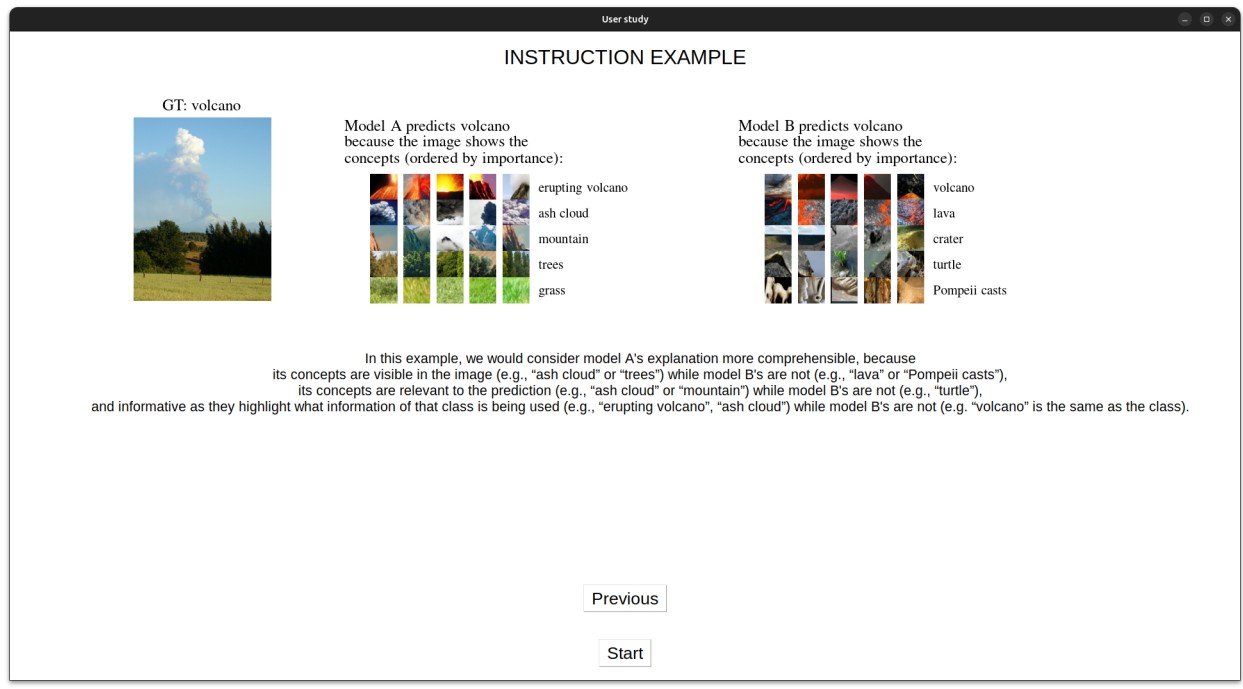

Figure 22: **Instruction example.**

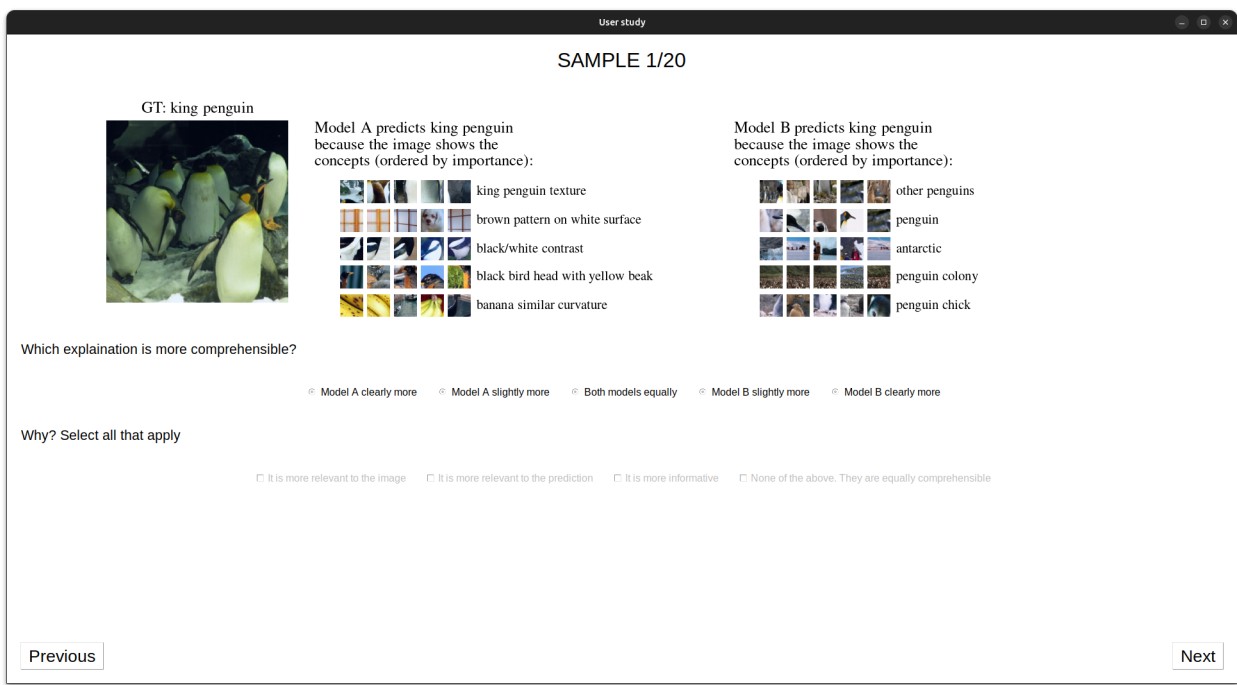

Figure 23: **User study sample.**

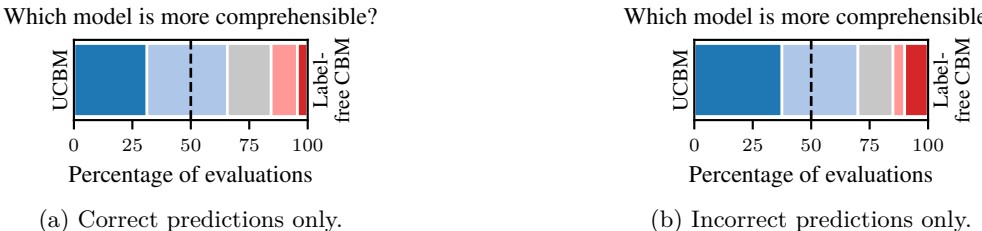

(a) Correct predictions only.

(b) Incorrect predictions only.

Figure 24: **Users strongly preferred UCBM with TopK concept selector over Label-free CBM for correct as well as incorrect predictions.**

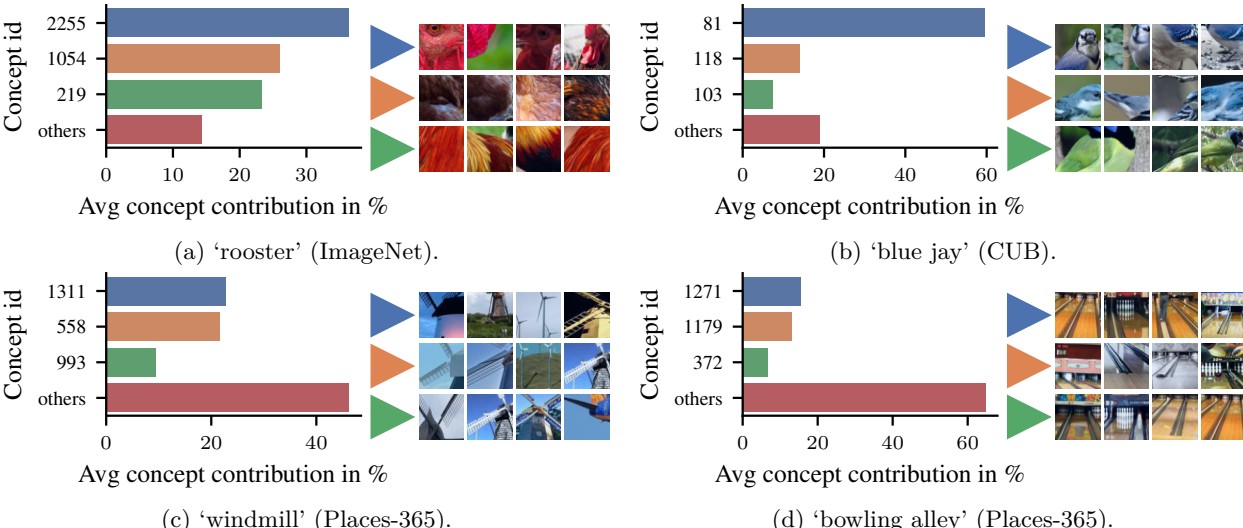

(a) 'rooster' (ImageNet).

(b) 'blue jay' (CUB).

(c) 'windmill' (Places-365).

(d) 'bowling alley' (Places-365).

Figure 25: **Visualization of decision rules learned by UCBM with TopK concept selector** on ImageNet (a), CUB (b) and Places-365 (c, d).

# K  Concept labeling with a large vision-language model

As an alternative to providing the top-activating image crops and manual concept labelling, we also experimented with large vision-language models (GPT-4o (Achiam et al., 2023)) to automatically label concepts. We prompted it with the top-9 image crops and task description:

*The nine pictures within the image are matching a specific concept.*
*Can you describe the concept with very few words (ca. 1–3)?*

Figure 26 shows the outputted concept labels for twelve, diverse concepts. Overall, we found that concept labels are mostly matching to the top image crops, e.g., Figures 26a, 26d, 26e and 26k. However, there are also concepts that may not be correctly labelled. For example, the large vision-language model outputs "motorcycle racing" for the image crops in Figure 26b. While this matches well with most of the image crops, it does not for the baseball player (bottom middle) and cyclist (bottom right). We suspect that the concept is representing a more general concept for "safety equipment" instead. For another example, in Figure 26h, the large vision-language model labelled the concept as "ocean textures". However, the image crops more likely resemble a starry sky rather than some ocean textures due to the point structure.

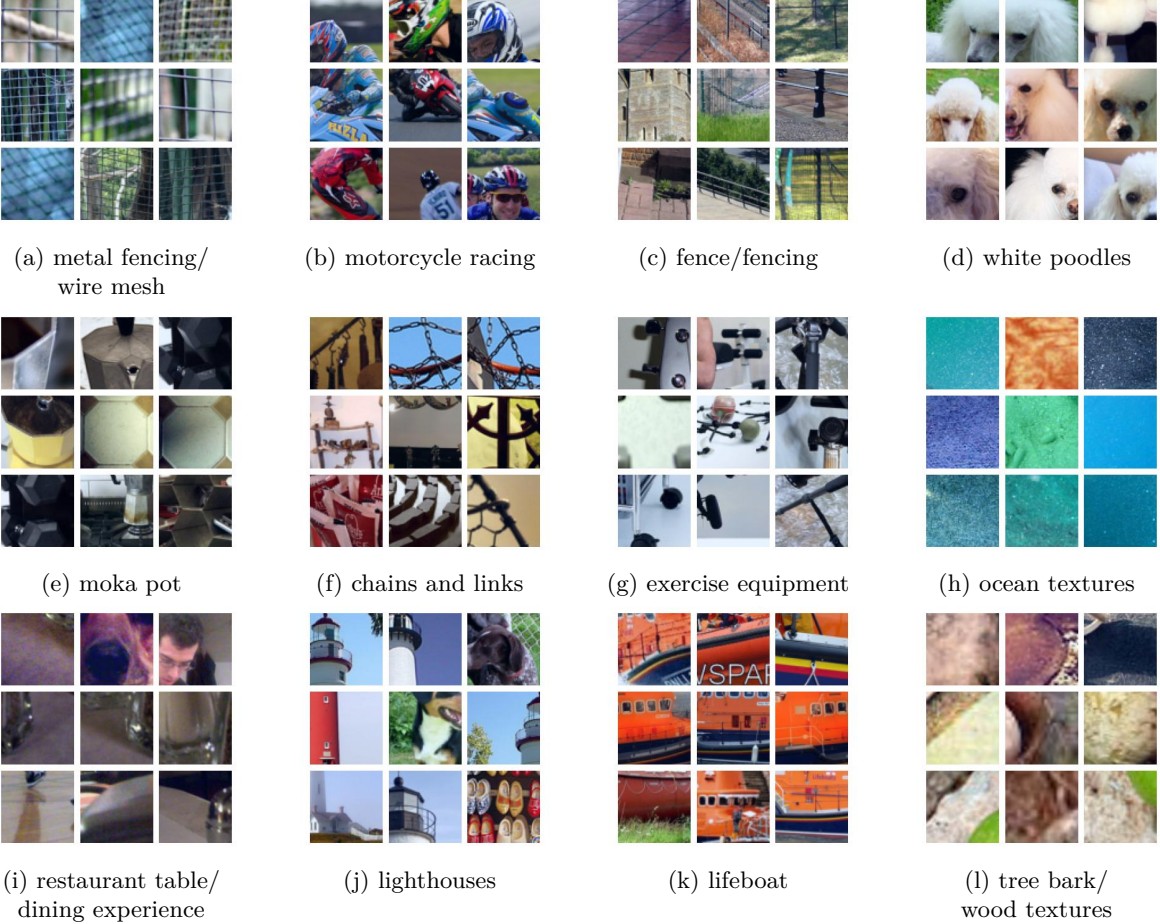

(a) metal fencing/
wire mesh

(b) motorcycle racing

(c) fence/fencing

(d) white poodles

(e) moka pot

(f) chains and links

(g) exercise equipment

(h) ocean textures

(i) restaurant table/
dining experience

(j) lighthouses

(k) lifeboat

(l) tree bark/
wood textures

Figure 26: **Labeling of concepts using large vision-language models.** The subfigures' captions are the labeling/descriptions that the large vision-language model (GPT-4o (Achiam et al., 2023)) assigned to the provided concept visualizations.

## L   Applications of UCBMs beyond image classification

Recent work applied concept-based models to tabular data (Zarlenga et al., 2023) and language models (Ismail et al., 2023). Although our primary focus is image classification—the domain where concept-based models have been studied most extensively—UCBMs are also applicable to other domains. Specifically, we first need to find the concepts again. For example, sparse autoencoders have become a popular method for uncovering human-understandable concepts in LLMs. Once we have found these concepts, we can train an interpretable classifier, as described in Section 2.2.

## M   Example prompt to the large vision-language model

Figure 27 shows an example prompt to the large vision-language model for the misclassification from the lower, left subfigure in Figure 11. Figure 28 shows the corresponding output from the large vision-language model.

## N   Robustness, fairness, and shape vs. texture bias of UCBMs

In the following analysis of robustness, fairness, and shape vs. texture bias of UCBMs, we focus on the UCBM model with the TopK concept selector trained on ImageNet.

**How robust are UCBMs?**   We evaluated the out-of-distribution robustness of UCBMs using the dataset provided in the model-vs-human toolbox (Geirhos et al., 2021), with the corresponding codebase available at `https://github.com/bethgelab/model-vs-human`. This dataset includes twelve parametric image distortions, such as uniform noise, rotations, etc. Figure 29 shows that UCBMs exhibit robustness comparable to that of the original black-box model. This is expected, as UCBMs likely inherit the biases encoded in the frozen bottleneck features of the black-box model.

**How fair are UCBMs?**   Recent work has demonstrated significant disparities in class-wise accuracy—referred to as "image recognition unfairness"—even on balanced datasets like ImageNet (Cui et al., 2024). As shown by Figure 30, this unfairness is evident in both black-box models and also our UCBM. Specifically, UCBM achieves a test accuracy of 100% for the best-performing class ('ostrich') and only 20% for the worst-performing class ('laptop computer'). The black-box model (ResNetV2) shows a similar pattern, with 100% accuracy for the best-performing class ('ostrich') and just 16% for the worst-performing class ('laptop computer'). These results are consistent with Cui et al.'s hypothesis that the underlying representations (frozen bottleneck features of the black-box model), rather than the classifier itself, are the primary source of this unfairness.

**Are UCBMs more shape or texture biased?**   To investigate shape vs. texture bias, we used the shape-texture cue conflict dataset introduced by Geirhos et al. (2019), employing the associated codebase available at `https://github.com/bethgelab/model-vs-human`. Figure 31 shows that UCBMs exhibit a texture bias similar to that of the original black-box model. This is again expected, as UCBMs likely inherit the biases present in the frozen bottleneck features of the black-box model.

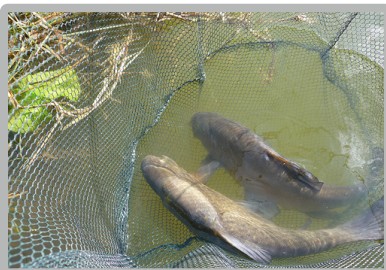

The image from above leads to the following output if we are putting it into a trained concept bottleneck model.

Ground truth: tench
Prediction: eel

Top-5 model prediction sorted by the logits:
Class eel: 7.71
Class tench: 7.05
Class silver salmon: 6.76
Class snoek fish: 6.08
Class sturgeon: 5.93

Largest five concept contributions (by absolute value) for ground truth class tench, sorted by size (and their weight in the final linear layer):
Concept 2088: 5.27 (0.34)
Concept 2987: 1.48 (0.20)
Concept 162: 0.17 (0.01)
Concept 2177: 0.01 (0.00)
Concept 2479: 0.01 (0.00)

Largest five concept contributions (by absolute value) for predicted class eel sorted by size (and their weight in the final linear layer):
Concept 333: 2.94 (0.34)
Concept 1367: 2.26 (0.20)
Concept 760: 0.72 (0.11)
Concept 2088: 0.68 (0.04)
Concept 2987: 0.51 (0.07)

Furthermore, the layer from the concept similarities to the prediction is a linear layer. Which weights (concept and class) should be adjusted by how much in order to get a correct classification for this image, while not changing any other classifications? Please give an answer in case of 'increase/decrease weight of class i and concept j by x'.

Figure 27: **Input prompt to the large vision-language model to edit the linear classifier's weights to correct an error.**

To get the model to correctly classify the image as "tench" instead of "eel," we'll need to adjust the weights in the final linear layer between the relevant concepts and the classes. Here's the approach to identifying which weights should be adjusted and by how much:

**Step-by-step Approach:**

1. Identify Misclassification and Concept Contributions:
   - The model currently predicts "eel" with a logit of 7.71, while "tench" has a logit of 7.05.
   - To change the classification to "tench," we need the logit for "tench" to be higher than the logit for "eel."

2. Calculate Difference Needed:
   - Currently, the difference between "eel" and "tench" logits is: $7.71 - 7.05 = 0.66$.
   - To correctly classify the image as "tench," the logit for "tench" must exceed "eel." Therefore, we need to adjust the weights such that the logit for "tench" increases by at least 0.66.

3. Determine Which Concept Weights to Adjust:
   - For "tench," the largest contributing concept is 2088 with a contribution value of 5.27 and a weight of 0.34.
   - For "eel," concept 2088 is also present with a smaller contribution (0.68) and a lower weight (0.04).
   - Adjusting concept 2088's weight for "tench" might be the most straightforward way to change the output because it has a significant contribution value.

4. Calculate Weight Adjustments:
   - To increase the "tench" logit by 0.66, we need to find an increase in the weight of concept 2088 for "tench":

     $$\text{Increase in logit} = \text{concept contribution} \times \text{weight increase}.$$

     Here, the concept contribution for 2088 is 5.27. So, we need to find the weight increase ($\Delta w$) such that:

     $$5.27 \times \Delta w = 0.66.$$

     Solving for $\Delta w$:

     $$\Delta w = \frac{0.66}{5.27} \approx 0.125$$

5. Adjust the Weight:
   - Increase the weight between "tench" and concept 2088 by approximately 0.125.

**Final Answer:**

   - Increase the weight of class tench and concept 2088 by 0.125.

This targeted adjustment will help correct the classification for this specific image without affecting other classifications significantly, as it focuses on the concept that most strongly contributes to the "tench" logit.

Figure 28: **Output of the large vision-language model to edit the linear classifier's weights to correct the error.**

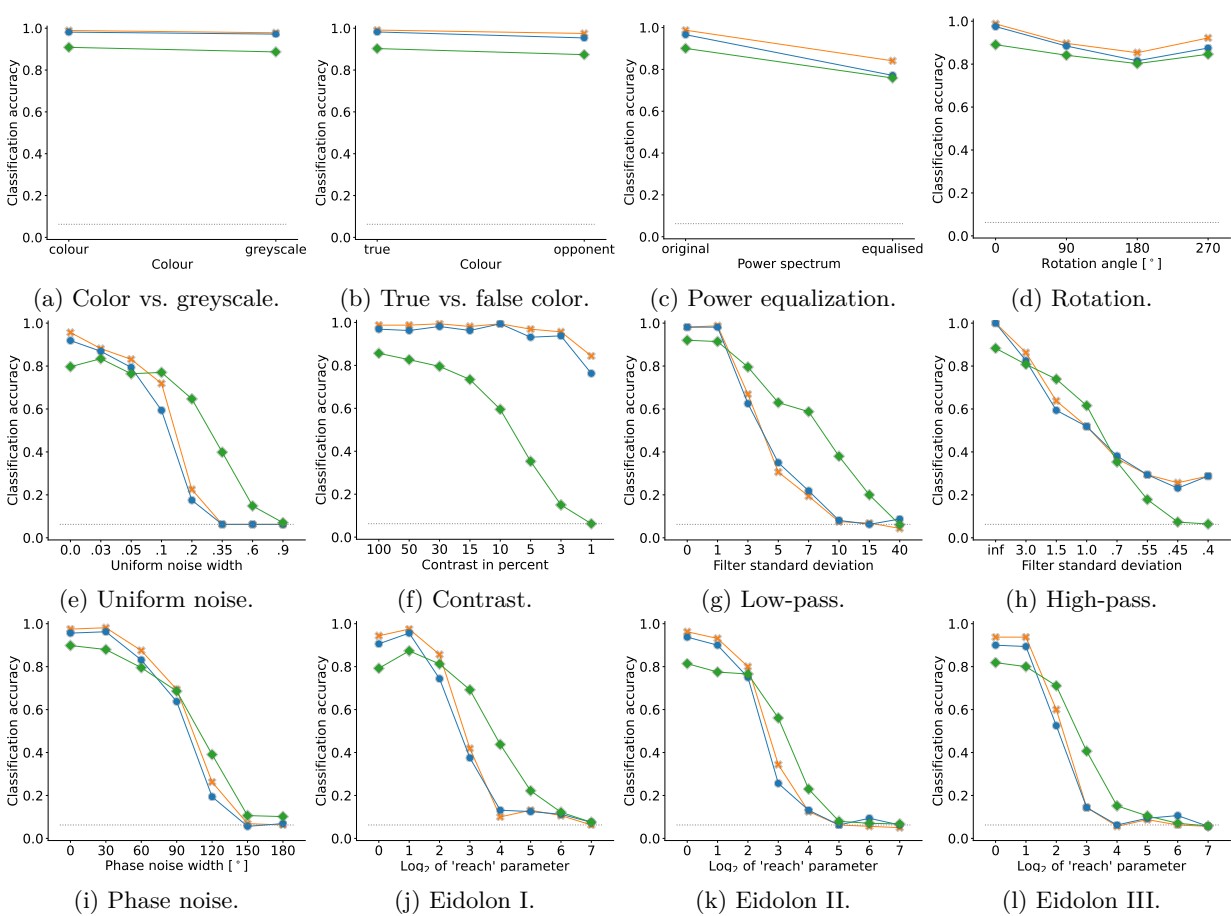

(a) Color vs. greyscale.  (b) True vs. false color.  (c) Power equalization.  (d) Rotation.

(e) Uniform noise.  (f) Contrast.  (g) Low-pass.  (h) High-pass.

(i) Phase noise.  (j) Eidolon I.  (k) Eidolon II.  (l) Eidolon III.

Figure 29: **Out-of-distribution accuracies for UCBM, the original black-box model (ResNetV2), and human observers.** UCBM behaves similar to the original, black-box model.

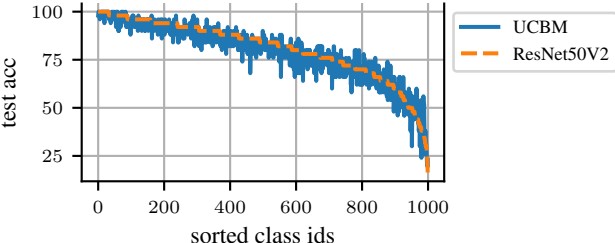

Figure 30: **Class-wise ImageNet test accuracy UCBM and ResNet50V2.** UCBM exhibits significant disparities in class-wise accuracy, indicating fairness issues similar to those of the original, black-box model. Class indices are sorted by the test accuracies of ResNetV2.

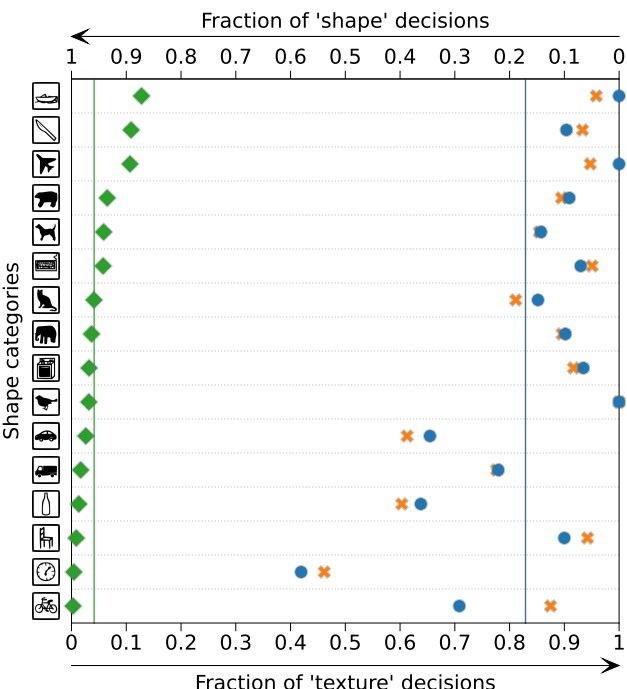

Figure 31: **UCBMs (blue circles) exhibit a texture bias similar to that of the original black-box model (ResNetV2, orange crosses).** In contrast, humans (green diamonds) are more shape biased.

