# OpenReview forum: "Selective Concept Bottleneck Models Without Predefined Concepts"
_TMLR — Accepted by TMLR_

### Review · Reviewer_7RW2 · 2025-02-28

**Summary Of Contributions:**

The paper advances the training of concept bottleneck models (CBM) by removing the need of knowing the concepts a priori, building on top of recent work in unsupervised concept discovery based on matrix factorisation.
The main novelty is in the introduction of non-negative matrix factorisation combined with a sparsity constraint to create a bottleneck layer that is then used to create the explanations.

The authors validate their approach against other CBMs approaches on diverse computer vision datasets and convolutional architectures (ResNet 18 and 50).  The evaluation is based on the model performance against other CBM approaches and on several insights on the discovered concepts, their relevance and understandability.
A user study is also included in the results.
The results show that the loss of performance given by their approach is minimal when it happens, and that in most cases their method grants higher accuracy than other approaches, and higher robustness of the discovered concepts.
Finally, they prove that their method also benefits from the other perks of CBMs such as generating model interventions.

**Audience:**

Yes

**Broader Impact Concerns:**

No need for broader impact statement

**Claims And Evidence:**

Yes

**Requested Changes:**

Some changes on the structure:
- I find it confusing to have a high-level explanation of the method in the introduction, because it is hard to follow at the first read of the paper. I would suggest reducing the level of detail on the methodology and focus on contextualising the work, adding details on the motivation and main challenges of the approach as opposed to the potential impact.
- Figure 1 could be improved by highlighting the "Alignments score" and "input-dependent concept selection" blocks as the main introduced novelty
- Figures 3, 4 and 5 should be improved. They are way too small and the values barely comparable. The top of Figure 3 is cut and some points barely visible.
- Figure 8 has a similar issue, the concepts are barely visible. Better to show y6ygless or increase the image size.
- Table 1 values should be reported as a percentage, otherwise the comparison is hard.
- Out of the 42 concepts found for Imagenet in Table 1 (and the 64 for CUB and 162 for Places365), what is the impact of each concept to the final prediction? An ablation study of the concepts would create a distribution of concept importance that could be interesting to see.
- It is not clear what is K in Table 3. Please report it for each result if it differs, otherwise just specify the value.
-The related work section should be largely improved. The paper https://arxiv.org/abs/2111.09259 should be in the references because it is the one that originated the actual idea of unsupervised concept discovery through matrix factorisation, so this should be at least mentioned. Similarly, recent work on identifying concepts in transformers architectures should be referenced (e.g.  https://arxiv.org/pdf/2412.06639). I would also expand the use of factorisation as a means of controlling model output, see for example https://arxiv.org/pdf/2206.00048. A discussion on directions that may be found by craft that represent more than one concept could be interesting to see, both in the experiments and also in the related work by mentioning polysemanticity.

**Strengths And Weaknesses:**

I like the paper and I find it interesting but I consider its novelty incremental. The main contribution comes from the combination of existing and interesting approaches with a sparsity constraint. This demonstrates that CBMs can be built also when no concepts are known a priori, solving a practical issue.

Strengths:
- It solves a practical problem demonstrating the applicability of CBMs to a wider spectrum of problems, which could have high impact.
- The evaluation is performed on multiple computer vision datasets with consistent results
- The results are positive, showing the efficacy of the method as opposed to existing baselines

Weaknesses:
- One main weakness I see with the approach is that it is hard to claim full interpretability because there seem to be concepts that are non-visible but that still influence the prediction for the predicted class and other classes. This phenomenon could have been analysed more in depth to characterise the limitations of the approach correctly.
- The paper structure could be improved, together with the exposition of some concepts and figures. For these points, see the requested changes.
- An important lack is the application of the approach to transformers models. One example of it is only briefly discussed in the appendix. I find it misleading to not have the same discussion in the paper. Similarly, not testing on other convolutional architectures than ResNet is a limitation. The argument would be much stronger with results provided on ResNet, Inception and ViT within the same manuscript.
- No mention is given on polisemanticity, and on the possibility that some directions may represent more than one concept. This is observed for individual neurons (https://arxiv.org/abs/2210.01892) but also for directions obtained through linear decomposition (https://arxiv.org/pdf/2307.06913).
- The related work is also lacking of other important works that have explored concept-based explainability and alternative ways of finding concepts, e.g. https://arxiv.org/pdf/2412.06639.

---

> ### Author Response · Authors · 2025-03-11
>
> We thank the reviewer for the detailed review and constructive feedback. Below, we address remaining concerns and questions.
>
> > One main weakness I see with the approach is that it is hard to claim full interpretability because there seem to be concepts that are non-visible but that still influence the prediction for the predicted class and other classes. This phenomenon could have been analysed more in depth to characterise the limitations of the approach correctly.
>
> We now quantify how often a top-5 most contributing concept to the prediction is also visible in the image, comparing our method vs. methods from the literature (Label-free CBM, VLG-CBM). Please refer to Tab. 8 in the revised manuscript. We find that our method nearly always relies on visible concepts, whereas Label-free CBM and VLG-CBM often rely on non-visible concepts that are merely correlated with the predicted class. This is particularly evident in misclassifications (see bottom table in Tab. 8 and Figs. 18f-18i). That said, we acknowledge that our method still occasionally uses non-visible concepts (6 out of 55 in Tab. 8). Completely alleviating this is an important direction for future work.
>
> > The paper structure could be improved, together with the exposition of some concepts and figures
>
> * Introduction: We see this as a stylistic preference but are open to revising it if other reviewers or the action editor find it also necessary.
> * Fig. 1: We have emphasized the key blocks using italics and added references to relevant subsections. We also refined the caption to better highlight our contributions.
> * Fig. 3-5: We increased plot sizes. In Fig. 3, we also added zoom insets.
> * Fig. 8: We have left this figure unchanged for now, as enlarging it would require removing the second example due to page limits. We welcome feedback from the reviewer whether we should still enlarge the figure or retain it as-is.
> * Tab. 1: We switched to percentages and moved the original table to the appendix.
> * K in Tab. 3: We are unsure which table this refers to. In Tab. 3, we added the missing $k$. However, we suspected that the reviewer referred to the non-enumerated table in Fig. 11. Here, we added the value of $k$ in the caption.
>
> > An important lack is the application of the approach to transformers models. One example of it is only briefly discussed in the appendix. I find it misleading to not have the same discussion in the paper.
>
> Unfortunately, due to page limit constraints, we are unable to include the transformer experiment in the main text.
>
> >  Similarly, not testing on other convolutional architectures than ResNet is a limitation
>
> The ResNet backbone choice follows standard evaluation protocols in the literature, where ResNets are commonly used as feature backbones for testing post-hoc concept-based models.
>
> > A discussion on directions that may be found by craft that represent more than one concept could be interesting to see, both in the experiments and also in the related work by mentioning polysemanticity.
>
> We now discuss polysemanticity in Sec. 5. Additionally, we quantified the degree of polysemanticity by analyzing the top-9 most activating image patches for 100 randomly chosen concepts (see App. B). This study indicates that while most concepts (88) appear monosemantic, a fraction (12) still exhibits polysemanticitiy.
>
> > The related work is also lacking of other important work
>
> We have now incorporated the suggested references.
>
> > Out of the 42 concepts found for Imagenet in Table 1 (and the 64 for CUB and 162 for Places365), what is the impact of each concept to the final prediction?
>
> Could the reviewer please clarify what is meant here?
>
> Note that we provide global importance scores of concepts for particular classes in, e.g., Fig. 10. For local explanations (i.e., per-sample), we always provide the concepts’ contributions to the prediction, as in Fig. 8.

---

> > ### Comment · Reviewer_7RW2 · 2025-03-14
> > **Thank you for addressing my comments**
> >
> > Dear Authors,
> >
> > Thank you very much for addressing my comments!
> >
> > As for my last question, it referred about having more descriptive statistics on the concepts. For example, instead of reporting the average active concepts, one could provide a breakdown of how many concepts are needed to describe each class. It seems from figure 10 that for pineapple there are mainly two concepts that cover most of the contributions. It would be interesting to see similar statistics for all classes. One could represent, for example, the cumulative contribution of the first two or three concepts for each class as a bar plot. Alternatively, one could also compute the average number of concepts required to have a contribution coverage of at least 90% for all classes. Does this clarify my question? I just wonder if two or three concepts are sufficient to explain most of the classes or if there are some classes that, for example, require dozens or even hundreds of concepts.

---

> > > ### Author Response · Authors · 2025-03-15
> > > **Re: Thank you for addressing my comments**
> > >
> > > Thank you for your reply and clarification! As suggested, we have now added an analysis (Figure 14 in Appendix E) that shows the average number of concepts needed for each class. We find that there are differences in the number of concepts needed between classes. We hope this addresses your question.

---

### Review · Reviewer_3BPc · 2025-03-01

**Summary Of Contributions:**

This paper introduces Unsupervised Concept Bottleneck Models (UCBMs), a new class of concept bottleneck models (CBMs) that first extracts concepts in an unsupervised manner, via sparse dictionary learning with CRAFT, and uses a selector to linearly combine the concepts and provide label predictions. The authors thoroughly compare UCBMs against CBMs that leverage supervision coming from visual-language models, like CLIP or Grounding DINO. Overall, this paper goes in the direction of understanding how to create more transparent methods based on concept-level predictions when supervision on concepts is not available.

**Audience:**

Yes

**Claims And Evidence:**

Yes

**Requested Changes:**

Overall, the authors have conducted a complete experimental verification on UCBMs compared to other models.

The interpretability of the proposed method heavily relies on (1) models in use and (2) post-hoc extraction method. The authors should clarify this point and address the weaknesses. It would be interesting to investigate if the guarantees in [Leemann et al. (2023)] are satisfied and concept discovery is successful.

**Strengths And Weaknesses:**

## **Strenghts**

The paper is well-written and provides a clear presentation of key components of UCBMs. The experimental analysis covers several settings, including a comparison to state-of-the-art label-free CBMs, ablations on concept selection, a user study on interpretability of model explanations, and error correction with LLMs. UCBMs are a natural baseline when coming to lack of supervision on concepts, which is of interest in the field of weakly/language-supervised CBMs.

Some interesting aspects that the authors contribute to spot:
1) Label-free CBMs and VLG-CBMs tend to rely on a multitude of concepts when coming to predictions. This depends on the linear layer in use, which, despite being optimized to be sparse, does not make input-label predictions depend on those few relevant concepts. As a consequence, concept-based explanations depend on many concepts and are hard to understand. The idea of the selector is interesting and also addresses a form of concept leakage. This augments existing studies in that direction [Havasi et al. (2022), Marconato et al. (2022), Zhang et al. (2025)].
2) Concept-based explanations of UCBMs remain competitive to Label-free CBMs and are typically preferred in the user study. Moreover, it seems that current language-supervised CBMs suffer from poor concept semantics, associating irrelevant concepts to the decision making. E.g., in Figure 8, the prediction for "sport car" highly depends on concepts like "room for 7 or 8 passengers" for LF-CBMs, and "special occasion" for VLG-CBMs.

These two aspects combined reveal that UCBMs remain competitive to these state-of-the-art language-supervised CBMs.

## **Weaknesses**

UCBMs inherit the limitations of leveraging concepts obtained through unsupervised post-hoc methods. I agree with the authors that UCBMs can benefit "from future unsupervised concept discovery methods", but, whether these concepts are even interpretable is not clear and depends primarily on the model used to obtain such concepts.

1) The authors should be more explicit about this aspect instead of sidelining it by saying "assuming linearity of concepts, as per the superposition hypothesis" (Section 2.1).
Authors should be more precise on this point, as interpretability is the main focus of the work, and connect to guarantees of discovery [Leemann et al. (2023)] and failures [Bortolotti et al. (2024)].

2) It must also be clear how concept-naming is obtained, e.g. through CRAFT and featuring in Figure 8, as I believe it is possible to attribute a name to some of the concepts but not all in general. Indeed, unsupervised concepts are in the order of thousands in Imagenet and Places-365. This is a critical point that should be highlighted.

3) Other techniques are also connected to UCBMs like Anycbms [Dominici et al. (2024)] for CBM distillation and concept-interventions in black-box models [Laguna et al. (2024)] and deserve a discussion.

### **References**

[Havasi et al. (2022)] Addressing Leakage in Concept Bottleneck Models, NeurIPS \
[Marconato et al. (2022)] GlanceNets: Interpretable, Leak-proof Concept-based Models \
[Zhang et al. (2025)] The Decoupling Concept Bottleneck Model, IEEE \
[Leemann et al. (2023)] When are Post-hoc Conceptual Explanations Identifiable?, UAI \
[Bortolotti et al. (2024)] A Neuro-Symbolic Benchmark Suite for Concept Quality and Reasoning Shortcuts, NeurIPS \
[Dominici et al. (2024)] AnyCBMs: How to Turn Any Black Box into a Concept Bottleneck Model, xAI \
[Laguna et al. (2024)] Beyond Concept Bottleneck Models: How to Make Black Boxes Intervenable?, NeurIPS

---

> ### Author Response · Authors · 2025-03-11
>
> We thank the reviewer for the thorough review and constructive feedback. Below, we address the remaining concerns and questions.
>
> > Whether these concepts are even interpretable is not clear and depends primarily on the model used to obtain such concepts
>
> We agree that interpretability depends on the chosen discovery method. In our work, we assessed interpretability by testing concept faithfulness (e.g., see Fig. 2) and inspecting top-activating image crops. However, we acknowledge that this may still fall short of full interpretability. Thus, we added a discussion in Sec. 5 and believe this is an intriguing direction for future work to explore.
>
> Unfortunately, Leeman et al [1] did not provide identifiability results for non-negative matrix factorization. Nonetheless, we find this is a promising direction for future work and have incorporated it into the discussion in Sec. 5.
>
> > It must also be clear how concept-naming is obtained
>
> We initially auto-generated concept descriptions using GPT-4o (details in Appendix J). However, as noted in your review, auto-labeling is not always reliable. Therefore, we manually reviewed and refined concept namings as needed, and also created some descriptions purely by hand.
>
> We have added a sentence in the first paragraph of Sec. 5 to make this more explicit.
>
> > Other techniques are also connected to UCBMs
>
> We have incorporated the suggested references in the relevant sections (introduction, method, related work). Note that while we had already cited Laguna et al [2], their author order changed from the arXiv preprint to the NeurIPS publication. We have updated the reference accordingly.
>
> ---
>
> [1] Leemann, Tobias, et al. "When are post-hoc conceptual explanations identifiable?." UAI (2023).
>
> [2] Laguna, Sonia, et al. "Beyond concept bottleneck models: How to make black boxes intervenable?." NeurIPS (2024).

---

> > ### Comment · Reviewer_3BPc · 2025-03-13
> > **Reply to authors**
> >
> > Thank you for adding the interpretability discussion in a separate section. Overall, I find the adopted changes sufficient.
> >
> > Few notes, reading the other reviews:
> > * I agree that testing on other architectures is also beneficial. It is worth remarking these results in the experimental discussion (not just a link).
> > * Monosemanticity and interpretability go hand in hand and I highlight again that only [1] gives guarantees to avoiding polysemanticity. I appreciate the inclusion of Appendix B, but the results are a bit hard to believe in this form, and the authors could include one example of monosemantic and one of polysemantic concepts.

---

> > > ### Author Response · Authors · 2025-03-15
> > > **Re: Reply to authors**
> > >
> > > Thank you for your reply and feedback. Below, we address your comments on the other reviews.
> > >
> > > > I agree that testing on other architectures is also beneficial. It is worth remarking these results in the experimental discussion (not just a link).
> > >
> > > Following reviewer 7RW2’s suggestion, we have now added an experiment using the InceptionV3 feature encoder on ImageNet (Appendix F). The results are consistent with those obtained using ResNet feature encoders. Additionally, we have included a brief discussion of these results in the experimental discussion.
> > >
> > > > I appreciate the inclusion of Appendix B, but the results are a bit hard to believe in this form, and the authors could include one example of monosemantic and one of polysemantic concepts.
> > >
> > > As suggested, we have added examples of both monosemantic and polysemantic concepts in Appendix B. Additionally, we have uploaded the top-9 image crops for all 3000 concepts of the ResNet-backbone/ImageNet experiment, along with the evaluation code for polysemanticity, at https://anonymous.4open.science/r/evaluate_polysemanticity-02EE (instructions are available in the README). In case additional information is needed, it would be helpful to us that the reviewer clarifies what is meant by “results [in Appendix B] are a bit hard to believe in this form”.

---

### Review · Reviewer_cisK · 2025-03-06

**Summary Of Contributions:**

The manuscript presents a method for converting an image classification model into a concept bottlenecked image classification model. The latter maps from the final image embedding into a concept space and then uses these concept activations to predict class labels.

- Concepts are discovered using non-negative matrix factorization (CRAFT Fel et al., 2023b).
- A sparse subset of the active concepts are selected as an extra step before computing class logits.
- Strong performance is reported for a very few active concepts per image.
- GPT-4o is used to update concept specific weights to correct misclassifications.

**Audience:**

Yes

**Broader Impact Concerns:**

The broader impact section claims no particular impact specific to the manuscript's methods beyond the general impact of advancing machine learning, but does not provide any evidence in this regard.
- Are UCBMs more or less robust than the black-box baseline?
- How do UCBMs do with regards to fairness metrics?
- Do UCBMs have texture or shape bias?

**Claims And Evidence:**

Yes

**Requested Changes:**

Please clarify the following:
- How exactly are top patches per discovered concept identified? The mapping from image representation to concept activations is done after the global average pooling, so there shouldn't be a patch level assignment if I understood correctly.
- The plots are very small making them hard to read. The x-axis label could be removed to make the plots taller perhaps?

**Strengths And Weaknesses:**

### Strengths

- Users prefer the proposed method over Label-free Concept bottlenecked baseline.
- Very few concepts are effect the classification logits because of the top-k selection.
- The proposed model is further bridging the gap between interpretable methods and the non-interpretable baseline.


### Weaknesses
- Lots of coefficients in the loss function, namely $\lambda_w, \alpha$, and $K$ for top-K, make this somewhat tricky to use in practice. One also needs to do concept dropout on top to avoid the degenerate scenario of using a single concept.
- It is unclear how this technique would be applicable to models that do object detection, point tracking, multimodal LLMs etc.

---

> ### Author Response · Authors · 2025-03-11
>
> We thank the reviewer for taking the time to review our paper and provide constructive feedback. Below, we address the remaining concerns and questions.
>
> > Lots of coefficients in the loss function
>
> The coefficients $\lambda_w$ and $\alpha$ are also used in previous state-of-the-art Concept Bottleneck Models (CBMs), such as Post-hoc CBMs (Eq. 1 in [1]), Label-free CBMs (Eq. 2 in [2]), or VLG-CBM (Eq. 7 in [3]). Our approach introduces only one additional hyperparameter: either $k$ (for Top-K) or $\lambda_\pi$ (for the other concept selectors). Notably, $k$ in Top-K is highly interpretable and can be easily chosen based on a user-specified target sparsity level (e.g., see Fig. 6 and App. C on how we set the coefficients).
>
> Regarding concept dropout: This is an orthogonal contribution that prior methods would also likely benefit from. For instance, VLG-CBM tends to rely heavily on just 1-2 concepts (see Figs. 8 & 18).
>
> > Applicability to object detection, point tracking, multimodal LLMs etc.
>
> Concept-based models are primarily studied in image classification, which is also our focus. However, recent work has extended them to other domains, including tabular data [4] and language models [5].
>
> To apply our approach, we would need to extract concepts using some unsupervised concept discovery method from a trained opaque black-box model. For (multimodal) LLMs, sparse autoencoders (SAEs) have recently become a popular approach to find human-interpretable concepts. Once we have created this concept bottleneck, we can learn a label predictor, similarly as done in our work or in [4,5].
>
> > How exactly are top patches per discovered concept identified?
>
> We first patchify the image (patch size = 64), compute feature activations from the bottleneck layer for each resized patch, and select the top-n patches with the highest cosine similarity to the concept.
>
> We have added a brief description in the “Quality of the discovered concepts” paragraph before Sec. 3.1.
>
> > The plots are very small making them hard to read.
>
> We have increased the sizes of Figs. 3-5, as also suggested by reviewer 7RW2, and added zoom insets to Fig. 3.
>
> For Fig. 8, increasing its size would require removing the second example due to page limit constraints. We would appreciate the reviewer’s feedback on whether we should enlarge the figure and drop one of the examples, or retain both examples.
>
> > The broader impact section claims no particular impact specific to the manuscript's methods beyond the general impact of advancing machine learning, but does not provide any evidence in this regard.
>
> We have refined the broader impact statement by specifying that our work advances the field of *concept-based models* rather than machine learning in general.
>
> Regarding the reviewer’s specific questions: Since our approach or any other post-hoc CBM approach, relies on the frozen bottleneck features, they likely inherit their biases (e.g., texture bias) and non-robustness. However, we would like to stress this is not the main goal of post-hoc CBMs. Instead, post-hoc CBMs aim to convert a previously opaque model into a more interpretable one. Nonetheless, we believe these are interesting directions for future work.
>
> —
>
> [1] Yuksekgonul, Mert, Maggie Wang, and James Zou. "Post-hoc concept bottleneck models." ICLR (2023).
>
> [2] Oikarinen, Tuomas, et al. "Label-free concept bottleneck models." ICLR (2023).
>
> [3] Srivastava, Divyansh, Ge Yan, and Lily Weng. "Vlg-cbm: Training concept bottleneck models with vision-language guidance." NeurIPS (2024).
>
> [4] Espinosa Zarlenga, Mateo, et al. "Tabcbm: Concept-based interpretable neural networks for tabular data." TMLR (2024).
>
> [5] Ismail, Aya Abdelsalam, et al. "Concept Bottleneck Language Models For protein design." ICLR (2025).

---

### Decision · Action_Editor_uNUQ · 2025-04-16

**Recommendation:** Accept with minor revision

**Comment:**

All reviewers agreed that this is a solid paper; all complaints are reasonably minor.  Some have not been explicitly addressed in the rebuttal (see reviewer cisK) and I encourage the authors to update the paper according to **all** feedback.

Major reasons for acceptance include novelty; improvements over the state of the art, supported by a rather extensive experimental analysis and a user study; and significance, considering the prominence of concept-bottleneck models among current deep learning and XAI trends.

**Audience:**

All reviewers agree that the contribution is closely aligned with TMLR's topics - namely deep learning, representation learning, and explainable AI - and I agree with them.

**Claims And Evidence:**

The reviewers generally agree that the experimental evidence (this is chiefly an empirical paper) in support of the proposed approach is sufficiently extensive and positive (it includes also a user study), and that the method itself compares well to the SOTA.  The paper does a good job at motivating and explaining the method itself.